# Assessment of responses of North Atlantic winter SST to the NAO on the interannual scale in 13 CMIP5 models

Yujie Jing[1,3], Yangchun Li[1,2,3], Yongfu Xu[1,2,3]

[1]State Key Laboratory of Atmospheric Boundary Layer Physics and Atmospheric Chemistry, Institute of

Atmospheric Physics, Chinese Academy of Sciences, Beijing 100029, China

[2]Laboratory for Regional Oceanography and Numerical Modeling, Qingdao National Laboratory for

Marine Science and Technology, Qingdao 266237, China

[3]Department of Atmospheric Chemistry and Environmental Sciences, College of Earth and Planetary

Sciences, University of Chinese Academy of Sciences, Beijing 100049, China

Correspondence: Yangchun Li (lych@mail.iap.ac.cn)

**Abstract.** This study evaluates the response of winter-averaged sea surface temperature (SST) to the winter North

Atlantic Oscillation (NAO) simulated by 13 CMIP5 Earth System Models in the North Atlantic (NA) (0-65°N) on the

interannual scale. Most of the models can reproduce an observed tripolar pattern of the response of the SST anomalies

to the NAO on the interannual scale. The model bias is mainly reflected in the locations of the negative response centers

in the subpolar NA (45-65°N), which is mainly caused by the bias of the response of the SST anomalies to the NAO-

driven turbulent heat flux (THF) anomalies. Although the influence of the sensible and latent heat fluxes (SHF / LHF)

on the SST is similar with each other, it seems that the SHF may play a larger role in the response of the SST to the NAO, and the weak negative response of the SST anomalies to the NAO-driven LHF anomalies is mainly caused by the overestimated oceanic role in the interaction of the LHF and SST. Besides the THF, some other factors which may impact the relationship of the NAO and SST are discussed. The relationship of the NAO and SST is basically not affected by the heat meridional advection transports on the interannual time scale, but it may be influenced by the cutoffs of data filtering, the initial fields and external-forcing data in some individual models, and in the tropical NA it can also be affected by the different definitions of the NAO indices.

**Keywords**: NAO; SST; turbulent heat flux; meridional advection; interannual scale

## 1. Introduction

There is a strong inverse relationship between Iceland's and the Azores' monthly mean sea level pressure (most significant in winter) in the North Atlantic (NA), which is called the North Atlantic Oscillation (NAO) (Walker, 1924). Studies have shown that the NAO has a significant impact on climate change in the Northern Hemisphere, including the significant impact on temperature and precipitation in Europe and the NA (Trigo et al., 2002). Because of the internal atmospheric dynamic process, the NAO is closely related to the location and intensity of storm track in the NA (Rivière and Orlanski, 2007). In addition, the NAO not only impacts the atmospheric field, but also the oceanic field through air-sea interactions, such as the sea surface temperature (SST) in the NA.

The influence of the atmospheric anomalies on SST is mainly reflected in the change of sea surface heat flux driven

by the change of local wind stress in the NA (Chen et al., 2015; Han et al., 2016), and this mechanism mainly occurs on

the interannual scale (Eden and Jung, 2001). Many studies have pointed out that the tripolar pattern of the SST anomalies

in the NA was driven by turbulent heat flux anomalies (sensible and latent heat fluxes, SHF and LHF) associated with

the NAO (Cayan, 1992; Marshall and Gareth, 2003; Visbeck et al., 2003; Deser et al., 2010). During the positive phase

of the NAO, the westerly winds in the subpolar NA and the northeast trade winds in the tropical NA are strengthened,

which causes the increased turbulent heat flux from the ocean to the atmosphere, while in the middle latitudes of the NA

wind speeds are weakened, which causes the reduced turbulent heat flux out of the ocean (Zhou et al., 2006, Deser et al.,

2010). After the positive phase of the NAO, some studies based on models suggest that, the Atlantic meridional

overturning circulation (AMOC) is intensified, and the strengthened meridional heat transport associated with enhanced

AMOC leads to broad scale SST warming (Sun et al., 2015, Delworth et al., 2017). Compared with other seasons, this

phenomenon is more obvious in winter (Flatau et al., 2003; Bellucci et al., 2006), and probably occurs on the interdecadal

and multidecadal scales (Eden and Jung, 2001, Gastineau et al., 2012). It should be noted that because there is a lack of

long-time AMOC observations and the AMOC plays a more active influence on the change of SST on the long timescale

(interdecadal and multidecadal scales), observational studies have not successfully linked the SST changes to the AMOC

variability (Buckley and Marshall 2016). In addition, SST anomalies also have feedbacks on the atmosphere, and the

dominant heat flux forcing to the NAO is associated with the later summer horseshoe SST forcing (Wen et al., 2005).

Furthermore, temperature anomalies of deeper seawater can also generate heat flux forcing to the atmosphere on the long

timescales (Yulaeva et al., 2001; Sutton and Mathieu, 2002).

Due to the time and space limitations of observations, many models are used to study the NAO. For example, Stoner et al. (2009) have evaluated the winter NAO simulated by coupled atmosphere-ocean general circulation models (AOGCMs), and pointed out that the spatial pattern of the NAO is more reasonable, but the action center of high pressure is west of the observation. In addition, Woollings et al. (2014) have simulated the mechanism of change of the NAO with an atmospheric circulation model (HiGEM), and proposed the impact of jets in the upper troposphere on the change of the NAO. The Coupled Model Intercomparison Project Phase 5 (CMIP5) (Taylor et al., 2012) includes more Earth system models with higher spatial resolution, which helps to better understand ocean and atmospheric variability and their interaction. The identification of the CMIP5 Earth System Models bias is important for the improvement of these models and development of climate projection. (Wang et al., 2014a, Wang et al., 2014b). For example, Liu et al. (2013) evaluated the SST variability in the NA warm pool simulated by 19 CMIP5 models, and considered that the bias of the radiation balance caused by the CMIP5 models' unreasonable simulation of high-level cloud fraction can impact the SST variability. Meanwhile, Wang et al. (2014b) evaluated the global annual mean SST simulated by the CMIP5 models and found that the SST in the Northern Hemisphere, especially in the NA, is underestimated. Wang et al. (2014b) also pointed that the underestimated SST mainly caused by the weaker AMOC and shallower AMOC cell compared to the observations. Wang et al. (2017) paid attention to the ability of the CMIP5 models to simulate annual NAO and found that basically all models can reasonably reproduce the spatial distribution of the NAO. So far, the relationship between the SST and NAO in the North Atlantic from the CMIP5 models has not been systematically evaluated, but it is of great significance to study the North Atlantic variability and climate change in the entire Northern Hemisphere. Multiple

observation-based studies have indicated that there is a close connection and strong interaction between the NAO and the tripolar pattern of winter SST anomalies (Czaja and Frankignoul, 2002; Chen et al., 2015). Therefore, the purpose of this paper is to evaluate whether the CMIP5 models can simulate the relationship between the NAO and SST in winter in the NA (0-65°N), to investigate the mechanism of the response of the SST to the NAO, and to explore the bias of models in simulating the response mechanism of the SST to the NAO.

## 2. Data and methods

### 2.1 Data

The observation-based data in this study are monthly sea level pressure (SLP) from 1948 to 2020 from the reanalysis dataset of the NCEP Reanalysis Derived Products (https://psl.noaa.gov/data/gridded/data.ncep.reanalysis.derived.html, Kalnay et al., 1996), and the monthly SST data from 1870 to 2016, which was produced by the Hadley Centre Global Sea Ice and Sea Surface Temperature (HadISST, climatedataguide.ucar.edu/climate-data/sst-data-hadisst-v11, Rayner et al., 2003). The 10-m wind speed (vm10) data from 1836 to 2015 used here are based on a synthesis of NOAA-CIRES-DOE Twentieth Century Reanalysis (V3) monthly average meridional and zonal wind from 1948 to 2018 (https://www.esrl.noaa.gov/psd/data/gridded/data.20thC_ReanV3.monolevel.html, Compo et al., 2011). The monthly sensible / latent heat flux data (SHF / LHF) from 1870 to 2016 used here were produced by NOAA-CIRES 20th Century Reanalysis version 2 (https://www.esrl.noaa.gov/psd/data/gridded/data.20thC_ReanV2.html, Compo et al., 2011). In order to verify the reliability of the reanalysis data, three other observation-based SHF / LHF data are selected for

comparative analysis, they are the NCEP/DOE AMIP-II Reanalysis Dataset (NCEP/DOE,

https://www.psl.noaa.gov/data/gridded/data.ncep.reanalysis2.html, Kanamitsu et al., 2002), Medium-Range Weather

Forecasts (ECWMF) interim reanalysis Dataset (ERA-interim, https://apps.ecmwf.int/datasets/data/interim-full-

mnth/levtype=sfc/, Dee et al., 2011), and Objectively Analyzed Air–Sea Fluxes (OAFlux,

ftp://ftp.whoi.edu/pub/science/oaflux/data_v3/monthly/turbulence/, Yu and Weller, 2007), respectively. The sea-surface

meridional velocity data from 1981 to 2017 are also used for analysis, which was produced by the Global Ocean Data

Assimilation System (GODAS, https://www.esrl.noaa.gov/psd/data/gridded/data.godas.html, Behringer and Xue, 2004).

The 13 Earth System Models used for this work are from the historical experiment (r1i1p1) of CMIP5 which is

integrated from the spinup results of the pre-industrial control experiment (piControl) and forced by the historical forcing

data after the Industrial Resolution (Table 1, cera-www.dkrz.de, Taylor et al., 2012). The simulated results from these

models provide the monthly average data of SLP, SST, SHF / LHF, and sea-surface meridional velocity (vo) during

1850–2005. There are 7 of these 13 models which conducted ensemble historical experiments, which started from

different integration times of the piControl, namely CanESM2, HadGEM2-CC, HadGEM2-ES, IPSL-CM5A-LR, IPSL-

CM5A-MR, MPI-ESM-LR, and MPI-ESM-MR. The experiments marked as r3i1p1 from these 7 models are adopted to

compare with those marked as r1i1p1, and the influence of initial fields on the relationship between the NAO and SST

is discussed. The results of piControl experiments are also used to investigate the historical forcing influence. In order

to make comparisons and analyses between the simulated and observed results, all variables are interpolated into a spatial

resolution 1°×1° by linear interpolation, and the time range of all variables from observations and models is 1965-2015

and 1955-2005 (except for the vo), respectively.

## 2.2 Methods

The anomalies of variables in the study are obtained by removing the trend from the seasonal mean data with the

least square method. The standardized variable is obtained by dividing the variable anomalies by their standard deviation.

The regression and covariance are performed with the standardized variables, for which the regression is conducted at

each grid.

The site-based observation-based NAO index is the difference of standardized sea level pressure between Lisbon,

Portugal and Stykkisholmur / Reykjavik, Iceland since 1864 (NCAR; www.climatedataguide.ucar.edu/climate-

data/hurrell-north-atlantic-oscillation-nao-index-station-based, Hurrell and Deser, 2009). Because the locations of the

NAO action centers are different between the model and the observation, and between the different models, we use the

method proposed by Zheng et al. (2013) to define the NAO index based on the results of the models. The winter NAO

index is defined as the difference in the standardized SLP, zonally averaged over the North Atlantic sector (30-80°N;

80°W–30°E), between the two latitudes that have the strongest negative correlation in SLP variability. The observed SLP

is used to verify the reliability of this NAO definition method: the correlation coefficient of the site-based NAO index

obtained by NCAR and the NAO index calculated by the method of Zheng et al.(2013) is 0.91, reaching statistical

significance at the 95% confidence level

The winter duration used to define the winter NAO indices / SST / wind speed / turbulent heat flux is December-

January-February (DJF). For the variables of DJF, the January in the given year is used as the reference to obtain the

winter variables. In other words, the variables for DJF of 1980 are obtained based on data in December of 1979 and

January and February of 1980. For the winter variables, when calculating the seasonal average NAO indices (DJF), the

winter season average of SLP is firstly calculated, and then the NAO indices are obtained.

Because the main cycles characterized by the interannual and decadal signs of the NAO are within 2-6 years and

above 8 years, respectively (Jing et al., 2019), the interannual scale is extracted using a 2-6 year Lanczos band-pass filter.

For the regression analysis between the NAO and ocean physical variables, the effective degree of freedom (DOF) is

calculated following Bretherton et al. (1999):

$$DOF = N(1\text{-r1r2})(1\text{+r1r2}) \tag{1}$$

Where N is the sample size, r1 and r2 are the lag-one autocorrelation coefficients of the time series of the two variables,

respectively.

The least-squares method is used to obtain the linear regression equation (y = ax + b), and with this method the

standardized regression coefficients of different variables are calculated. The NAO-driven SHF / LHF / sea-surface

meridional velocity anomalies are extracted by using the regression of these variables against the NAO indices.

In order to understand the mechanisms of the impact of the NAO on the SST, we need to know the main factors

leading to the change of the SST. The variability of the SST is described by:


$$C_0 \frac{\partial SST'}{\partial t} = Q' + A' = Q_R' + Q_B' + A' \tag{2}$$

$$Q_B = -Q_S - Q_L \tag{3}$$

Here, $C_0$ is the thermal capacity of the upper mixed layer of the ocean, which is approximately constant, A is the divergence of ocean heat transport, Q is the air–sea heat flux and has both radiative ($Q_R$) and turbulent ($Q_B$) components. The turbulent heat fluxes are the sensible ($Q_S$, SHF) and latent ($Q_L$, LHF) heat fluxes (the positive value indicates the flux from the sea surface to the atmosphere). Among them, the SHF and LHF are mainly related to wind speed (U) and SST, which are usually calculated by the following equations:

$$Q_s = \rho C_P C_S (SST - T_a)|\varDelta \vec{U}| \tag{4}$$

$$Q_L = \rho L_P C_L (q_s - q_a)|\varDelta \vec{U}| \tag{5}$$

where $\rho$ is a near surface air density, $C_p$ is the specific heat of the air, $L_p$ is the latent heat of evaporation, $C_S$ and $C_L$ are the transfer coefficients of sensible and latent heat fluxes, respectively, $T_a$ is the temperature of the atmosphere near the sea surface, $\varDelta \vec{U} = \vec{U}_a - \vec{U}_s$ is the vector difference between the wind speed at the sea surface and the sea surface current speed, in which the current speed is often neglected, $q_a$ and $q_s$ correspond to saturation specific humidity of air over sea surface and sea surface temperature, respectively. $q_s$ is usually calculated by the saturation humidit $q_{sat}$, for pure water at SST:

$$q_s = 0.98 q_{sat}(SST) \tag{6}$$

where a multiplier factor of 0.98 is used to take into account reduction in vapor pressure caused by a typical salinity of 34 psu. The methods adopted by the observation-based products and models to calculate the SHF / LHF are similar, which are mainly based on the bulk formula, but may use different parameters, so the above equations (2-6) only help us to understand the relationship between the SST and SHF / LHF, which are not the exact formulas used in the

observation-based products and models.

The variations of the heat energy caused by the surface meridional velocity at each grid can be expressed by the

following formula:

$$\Delta Q/\Delta t = SST_S \times V_S \times dx_s \times dz \times \rho \times C_p - SST_N \times V_N \times dx_N \times dz \times \rho \times C_p \tag{7}$$

where $C_p$ is specific heat capacity of seawater, and the subscripts S and N are the SST at the southern and northern

boundary of the grid, respectively. It is assumed that the density and specific heat capacity of adjacent seawater are the

same, and the meridional variation of the SST caused by the surface meridional velocity at each grid can be expressed

by the following formula:

$$\Delta SST/\Delta t = \frac{\Delta Q/\Delta t}{\Delta V \times \rho \times Cp} \tag{8}$$

where $\Delta V$ is the volume of the grid.

## 3   Results

### 3.1 Simulated basic state of the winter NAO and SST

### 3.1.1   Space state

An empirical orthogonal function (EOF) analysis is performed on the standardized winter-averaged North Atlantic

sea level pressure to obtain the first mode (EOF1) of the sea level pressure (SLP) field, that is, the NAO mode (Hurrell

and Deser, 2009). Figure 1 shows the NAO modes of the observation and CMIP5 model simulations. The NAO mode

calculated with the observed SLP is significant, which explains 51.8% of the total variance. The explanation variance of

the NAO mode by the models ranges from 27.5% - 56.4%, and the explanation variance of most models is lower than

that of the observation-based result. It's worth noting that the NAO mode simulated by the HadGEM2-CC in 1955-2005

is not statistically significant. The observation shows that the low-pressure action center of the NAO is at around 77.5 °

N and 2.5 ° E, and that the high-pressure action center is around 42.5 ° N and 2.5° W (Fig. 1). The CMIP5 models can

basically reproduce the NAO mode, although there are some slight differences of the locations of the NAO action centers

between different models and between the models and observation. The differences between the NAO patterns simulated

by these models with the same external-forcing data are probably induced by their different model structures and values

of parameters. In addition, many studies based on the observation pointed out that there is a shift in the action centers of

the NAO (Jung et al. 2003; Moore et al., 2013) and this shift is related to the phase of the NAO (Cassou et al. 2004, Jing

et al., 2019). The locations of the NAO action centers simulated by most of the 13 CMIP5 models in different NAO

phases do not show obvious movements illustrated by the observation (Fig. S1), which means that the climate variations

simulated by the models are more symmetrical than the actual situation.

The CMIP5 models can basically reproduce the spatial distribution of the SST in the NA (spatial correlation

coefficients with the observations are all above 0.99, Fig. S2), although the CMIP5 models underestimate the annual

mean SST of the NA (Wang et al. 2014b). In terms of interannual variability of winter SST in the NA (0-65°N) (Fig. 2),

all CMIP5 models can reproduce the strong interannual variability of the SST in the Gulf Stream extension, but the

simulated strong interannual variability of the SST by most models is more easterly than the observations. With a climate

system model, Siqueira and Kirtman (2016) found that the change of ocean component model resolution can change the

simulated SST variability, locations of atmospheric circulation anomalies, and air-sea interactions in the North Atlantic.

The change is induced by the impact of the resolution on the ocean dynamics, such as ocean fronts and eddies in the

Gulf Stream which can be well resolved in the high resolution model with the horizontal resolution of $0.1°\times0.1°$.

Nevertheless, the highest horizontal resolution of these ocean component models used in this study is $0.4°\times0.4°$ (MPI-

ESM-MR), and the comparison of MPI-ESM-LR and MPI-ESM-MR, both of which are from the same institution and

with different ocean component model resolutions, shows that the SST variability in the Gulf Stream is not significantly

different. This indicates that the resolution of these models is still not enough to investigate the SST variability in the

Gulf Stream and may explain the deviation between the simulated SST variability and the observed one. In addition,

some models also simulate strong interannual variability at higher latitudes that is not observed.

### 3.1.2  Temporal period

Figure 3a shows the periods of the observation-based NAO indices provided by NCAR and model-based NAO

indices calculated with the method proposed by Zheng et al. (2013). The power spectra of the NAO indices are also

shown in Fig. S3. The significant periods (at a 90% confidence level) of the observed NAO index are 3, 4.8 and 8-10

years shown as the red line in Figure 3a, characterized by interannual and decadal signals. Most models can reproduce

significant interannual signals of around 3 years, and CESM1-BGC, GFDL-ESM2M, IPSL-CM5B-LR, MPI-ESM-MR

and NorESM1-ME can reproduce the interannual signal of 4-5 years. It should be noted that HadGEM2-CC without

significant EOF1 of SLP does not have a significant interannual period of the NAO index. Therefore, we will not emphasize the analysis of simulated results from HadGEM2-CC in the following content. Compared with the observation, the model bias of the NAO periods is mainly reflected on the decadal scale, which is consistent with the analysis of the CMIP5 models by Wang et al. (2017) based on the annual NAO. Only 5 models such as CanESM2, HadGEM2-CC, IPSL-CM5A-MR MPI-ESM-LR and MRI-ESM1 can reproduce the decadal signals longer than 8 years, and GFDL-ESM2G can simulate the periods of 16 and 18 years characterized decadal signal that are not observed.

Figures 3b and 3c show the observed and simulated periods of winter area-averaged SST anomalies in the subtropical (25-45°N) and subpolar NA (45-65°N), respectively. The power spectra of the area-averaged SST anomalies are also shown in Fig. S4 (for the subtropical NA) and S5 (for the subpolar NA). In the subtropical NA, the observed area-averaged SST anomalies have a significant interannual signal of 2 years. Most of CMIP5 models can reproduce the 2-4 year interannual signals of SST in this region. Some models such as HadGEM2-CC / ES and IPSL-CM5A-LR / MR can simulate the decadal signal of 8-20 years that is not observed, and some models even produce the multi-decadal signal, such as IPSL-CM5B-LR and NorESM1-ME. In the subpolar NA, the observed area-averaged SST anomalies have a significant interannual signal of 3.5 years. There are 6 models that reproduce a significant period of the area-averaged SST anomalies about 3.5 years, namely CanESM2, CESM1-BGC, GFDL-ESM2M, IPSL-CM5A-MR, IPSL-CM5B-LR, and MPI-ESM-LR, and except for GFDL-ESM2G most models can reproduce the 2-6 year interannual signal of the SST in the region. Some models such as CESM1-BGC, GFDL-ESM2G / M, HadGEM2-CC / ES, IPSL-CM5A / B-LR, and MRI-ESM1 can also simulate the decadal / muti-decadal signal of over 8 years that is not observed.

Based on the above analysis, simulated periods of the NAO indices on the interannual scale are more consistent

with the results of observations compared to those on the decadal scale. The observed periods of the area-averaged SST

in the subtropical and subpolar NA only present interannual signals. In addition, the impact of the atmospheric anomalies

(NAO) on the SST in the NA is mainly reflected in the impact of local change of wind stress on the sea-air heat flux on

the interannual scales (Eden and Jung, 2001; Chen et al., 2015; Han et al., 2016). Therefore, we will extract the

interannual signal of 2-6 years by band-pass filter based on the periods of the NAO and area-averaged SST anomalies to

evaluate the relationship between the simulated NAO and SST in the CMIP5 models on the interannual scale.

## 3.2 Responses of NA (0-65°N) SST to the NAO

Figure 4 shows the regression coefficients (RCs) of the winter-averaged SST anomalies against the NAO indices

on the interannual scale in the NA (0-65°N). The significant RCs between the observed NAO indices and SST anomalies

give out a tripole pattern along the meridional direction with positive RCs in the subtropical region (25-45°N) and

negative RCs in the both tropical (0-25°N) and subpolar regions (45-65°N), which is consistent with Walter and Graf

(2002) and Chen et al. (2015). Compared with the observation, most of the models can roughly reproduce the tripole

pattern of the response of the SST anomalies to the NAO. In the region around 20°N, all models can reproduce the

significant negative response (reaching a 95% confidence level) east of 40°W, and in the subtropical NA, 10 models can

reproduce significant positive response of the SST anomalies to the NAO near the American coast. The main difference

in the RCs between the modeled and observation-based results occurs in the subpolar region where the simulated

locations of the negative response centers by some of models are different from the observation-based results, especially

in CanESM2, HadGEM2-ES, IPSL-CM5A-MR, MPI-ESM-LR / MR, and NorESM1-MR. The simulated and observed

factors affecting the response of the SST to the NAO will been compared to explore the reasons for the different response

of the SST anomalies to the NAO mainly in subpolar NA.

250   In HadGEM2-ES, the low-pressure action centers of the NAO are slightly further south than observations (Fig.1,

the low-pressure action center is at around 67.5°N), and the negative response center of the SST to the NAO is also

further south than observations. However, in some models (for example IPSL-CM5A-MR and IPSL-CM5B-LR), even

if the locations of the low-pressure action center of the NAO are close to that of the observation or even further north,

the negative response center of the SST anomalies to the NAO is further south than the observation-based results.

Therefore, there must be other reasons that impact the relationship of the SST and NAO in the subpolar NA in some

models.

### 3.2.1  The role of wind speed

   Since the influence of the NAO on the SST is mainly through the wind field in the NA (Zhou et al., 2006; Deser et

al., 2010), in order to evaluate the mechanism of the influence of the simulated NAO on the SST in the NA, the response

of the wind speed to the NAO should be firstly considered. Figure 5 shows the RCs of the sea surface wind speed

anomalies against the NAO indices on the interannual scale, which clearly shows a meridional tripole pattern with

negative RCs in middle latitudes (30-40° N) and positive RCs in the both tropical and high latitudes (north of 40°N).

This distribution pattern is closely similar to the NAO-SST relationship, which is consistent with the results of Cayan (1992), Marshall and Gareth (2003), Visbeck et al. (2003) and Deser et al. (2010). All the CMIP5 models can reproduce the impact of the NAO on the sea surface wind field. During the positive phase of the NAO, wind speed is strengthened in the tropical NA and high latitudes, and weakened in middle latitudes. This is consistent with the fact that during the positive phase of the NAO, the deepening of the low pressure in Iceland causes the anomalous east wind superimposed on the mid-latitude westerly wind, which weakens the mid-latitude wind speed (Deser et al., 2010; Chen et al., 2015). It should be noted that in the subpolar NA, the locations of positive response center of wind speed anomalies to the NAO are consistent among different models, which indicates that the difference between the locations of the NAO low-pressure action center simulated by these models has little influence on the locations of response center of wind speed anomalies to the NAO.

According to Eqs. (2-5), the wind speed anomalies impact the SST by affecting the turbulent heat flux, but the wind speed only affects the magnitude of the turbulent heat flux. Therefore, when analyzing the effect of wind speed anomalies on the turbulent heat flux, it is necessary to consider the direction of the turbulent heat flux that is determined by the difference of temperature and specific humidity between the atmosphere and the sea surface. From the results of the multi-year averaged winter sensible / latent heat fluxes (SHF / LHF, Fig. S6), the observed and simulated SHF / LHF are all from the sea to the atmosphere. Considering the directions of the SHF / LHF, the increase in wind speed can significantly increase the turbulent heat flux transported from a large region of sea surface to the atmosphere.

In order to make sure the accuracy of the observed multi-year averaged winter SHF / LHF, three other observation-

based SHF/LHF data in winter are selected and all of these datasets are in the same periods from 1980 to 2015. The distributions of the SHF / LHF from the 4 reanalysis databases are generally consistent with each other, and the main difference among these datasets is in the intensity of high values, especially in the high value center of the LHF located in the tropical NA (Figure 6a). The response of the SHF / LHF anomalies to the NAO in these 4 datasets is also close to

each other, and the main difference among these datasets still occurs in the tropical NA (Figure 6b). Based on the above analysis, it can be concluded that the difference among the observation-based SHF / LHF does not affect the investigation of the relationship of the SHF / LHF and NAO / SST in this study because the regions of concern are mainly the subtropical and subpolar NA. In the following text, unless otherwise specified, the observation-based SHF / LHF is the data from the NOAA-CIRES 20th Century Reanalysis version 2.

Most models seem to reproduce the maximum SHF in the Labrador Sea and the large SHF in the Gulfstream Basin (Fig. S6), and the observed distributions of the LHF are also well reproduced in these models, except that some models such as CanESM2 and IPSL-CM5B-LR underestimate LHF in the subpolar NA (Fig. S6). However, there is no evidence that the simulation bias of the magnitude of the multi-year average LHF can affect the relationship between the NAO and SST anomalies.

Figure 7 shows the RCs of winter turbulent heat flux anomalies against the NAO indices. The significant observed RCs between the NAO indices and SHF / LHF anomalies in the NA (0-65°N) indicate a meridional tripole pattern with negative RCs in the subtropical region and positive RCs in the both tropical and subpolar regions. The simulated and observed locations of the positive RCs in the subpolar region are almost same, which further illustrates that the bias of

the location of the NAO low-pressure action center by CMIP5 models may have little influence on the NAO-SST

relationship. The spatial distribution of the observed RCs is consistent with the results of Eden and Jung (2001) and

Deser et al. (2010), and is generally consistent with the meridional distribution of the observed RCs of the sea surface

wind speed anomalies against the NAO, from which we can infer that the wind speed anomalies related to the NAO can

impact the SHF / LHF anomalies. During the positive phase of the NAO, the increase of wind speed in the tropical and

subpolar NA strengthens the turbulent heat flux transported from the ocean to the atmosphere, while the weakening of

the wind speed in the subtropical NA weakens the turbulent heat flux. The CMIP5 models can basically reproduce the

significant RCs of turbulent heat flux anomalies against the NAO, for which the meridional distribution pattern is the

same as the RCs of the wind speed anomalies against the NAO indices.

### 3.2.2    The role of SHF

The variability of the SHF and SST is related. According to the calculation formula of the SST and SHF, the increase

of the SHF can decrease SST (Eqs. 2-3), while the decreased SST can further decrease the SHF (Eq. 4). Therefore, when

the variations of the SST and SHF are negatively correlated, it can be inferred that the change of the SHF influences the

SST, which means that the atmosphere forces the ocean; when the variations of the SST and SHF are positively correlated,

the change of the SST leads to the change of SHF, which means that the ocean forces the atmosphere.

Figure 8a is the observation-based and simulated RCs of the winter-averaged SST anomalies to the NAO-driven

SHF anomalies obtained by the linear regression of the SHF against the NAO indices. As expected, the observed winter-

averaged SST anomalies show significant negative RCs against the observed NAO-driven SHF anomalies in the high

value centers of RCs (absolute value) of the SST anomalies against the NAO index, except for a little region in the

eastern NA around 20°N where the RCs of SST against the NAO-driven SHF is positive, which is consistent with that

the RCs of SHF/SST anomalies against the NAO index are negative in this region (Fig. 7a, Fig. 4). It indicates that the

response of the SST to the NAO is really related to the response of the SST to the NAO-driven SHF and the atmosphere

forces most regions of the North Atlantic Ocean in winter. Considering the significant relationship between the SST /

SHF / wind speed anomalies and the NAO, it can be concluded that in winter the NAO can impact the SST by affecting

the SHF in the most regions of the NA through the change of wind speed. There are some differences between the

modeled and observed relationships of the SST and SHF. In the subtropical NA, there are no negative significant response

centers of the SST anomalies to the NAO-driven SHF anomalies near the American coast in CESM1-BGC, IPSL-CM5A-

LR, and NorESM1-ME, so the three models cannot simulate positive response centers of the SST anomalies to the NAO.

The positive significant response of the SST anomalies to the NAO-driven SHF in the subtropical NA of IPSL-CM5A-

MR and NorESM1-ME also induces a negative significant response of the SST anomalies to the NAO. In the subpolar

NA, the locations and magnitude of negative response centers of the SST anomalies to the NAO-driven SHF anomalies

in some models are not consistent with the observation-based results, but are consistent with those of the SST anomalies

to the NAO by all CMIP5 models. This can partly explain why the negative response center of SST anomalies to the

NAO in the subpolar NA in some models are inconsistent with the observation.

There may be two reasons for the bias of the locations and magnitude of negative response centers of the winter-

averaged SST anomalies to the NAO-driven SHF anomalies by models: The areas where air-sea interaction is dominated

by the atmosphere are different from the observation; there may be other factors which play dominate role to the variation

of the SST and further impact the relationship between the anomalies of the SST and SHF. To investigate the reason,

lagged (leaded) covariance analysis of monthly anomalies of the SHF and SST is used and shown in Fig. 8b. The lagged

or leaded time is two months. Here, the monthly SHF anomalies from October to April and monthly SST anomalies from

December to February of the same year are used. When the observed SST anomalies lag (lead) SHF anomalies by 2

months the covariance between the SHF and SST anomalies is negative (positive) in most regions. When the change of

SHF synchronizes with the change of SST, the covariance between the anomalies of the SHF and the SST is negative in

the subpolar and western subtropical NA, which means that the forcing from the SHF / atmosphere to the SST / ocean is

still dominated in the interaction between the SHF and SST in these regions.

When the SST anomalies lags by 2 months onto SHF anomalies, all CMIP5 models can reproduce the negative

covariance between SHF and SST anomalies in the most regions of the NA, although there are some models that simulate

weak positive covariance in some regions of the subpolar NA, such as GFDL-ESM2M, HadGEM2-CC/ES, IPSL-

CM5A-L/MR, MPI-ESM-LR, and MRI-ESM1, indicating that other factors (such as the internal motion of ocean) have

an impact on the variations of the SST in the regions beyond the SHF in these models. When the change of SHF is

synchronized with the change of the SST, most models reproduce the weakening of the SHF's influence on the SST,

especially in the subtropical and tropical NA, and in above 6 models the geographical range of the negative covariance

in the subpolar NA is still smaller than that of the observed data, especially in MRI-ESM1. It is worth mentioning that

the locations of the covariance center simulated by most models in the subpolar NA are consistent with that of the RCs of the winter-averaged SST anomalies against the NAO-driven SHF anomalies and that of the winter-averaged SST anomalies against the NAO, while in some models, such as MRI-ESM1 and NorESM1-ME, the performance of simulating covariance of the SHF and the SST is not consistent with that of simulating response of the SST to the NAO-driven SHF. In MRI-ESM1, the covariance of the SHF and the SST in the subpolar NA deviates from the observation-based results, while the response of the SST to the NAO-driven SHF and NAO is close to the observation results. In NorESM1-ME, the covariance of the SHF and the SST is close to the observation-based results in the subpolar NA, while the response of the SST to the NAO-driven SHF and NAO is biased in this region. These demonstrates that in some models there are other factors that can influent the relationship of the NAO-driven SHF / NAO and the SST in the subpolar NA.

### 3.2.3   The role of LHF

The LHF is calculated by wind speed and the difference between the saturation specific humidity of lower air and sea surface. Because the saturation specific humidity of sea surface is a function of the SST (Eq. 6), according to the calculation formulas of the SST and LHF (Eqs. 2-3, 5-6), the relationship between the LHF and SST is similar to the one between the SHF and SST. It means that when the variations of the SST and LHF are negatively correlated, the atmosphere forces the ocean through the LHF, and that when the variations of the SST and LHF are positively correlated, the ocean forces the atmosphere.

Figure 9a is the RCs of the observed and simulated winter-averaged SST anomalies against the NAO-driven LHF

anomalies. The distributions of the RCs are similar to those of the SST anomalies against NAO-driven SHF anomalies

in a large area of NA. The main difference between the response of the SST to the SHF and to the LHF is that the

observed and modeled positive RCs of the SST anomalies against NAO-driven SHF anomalies in the eastern NA around

20°N do not occur in the regression of the SST anomalies against NAO-driven LHF anomalies. It indicates that the

influence of the LHF on the SST probably controls the RCs of the SST anomalies against the NAO in this region. The

observed difference between the relationship of the SST and NAO-driven SHF and that of SST and NAO-driven LHF is

well reproduced by most models, and the main bias of the simulated response of the SST to the NAO-driven LHF is also

close to that to the NAO-driven SHF. In each model the locations and magnitude of the negative response centers of the

SST anomalies to the NAO-driven LHF anomalies in the subpolar NA are very similar with those to the NAO-driven

SHF anomalies. Therefore, the biases of the relationship between the SST anomalies and NAO-driven SHF / LHF

anomalies together lead to the bias of the locations and magnitude of the negative response center of the SST anomalies

against to the NAO in the subpolar NA in these models.

Figure 9b shows the lagged (leaded) covariance between the anomalies of the LHF and SST. As well as the

relationship of the SST and SHF, when the observed SST anomalies lag (lead) those of the LHF by 2 months, there is an

obviously negative (positive) covariance in a large region of NA, which indicates that the change of LHF (SST) can

influent the change of SST (LHF) after two months. It should be noted that whether the change of the LHF is ahead of

or synchronized with the SST the geographical ranges and magnitudes of the negative covariance of LHF and SST are

smaller than those of the SHF and SST.   For example, when the changes of the SST and LHF are synchronized, there

is an obviously positive covariance in the subtropical NA, which has a larger value and a greater range than that of the

synchronized SHF anomalies and SST anomalies. This demonstrates that the time-scale of the LHF affecting SST is

shorter than that of the SHF affecting SST, and the ocean plays an important role in the interaction of LHF and SST in

large region of the NA. The CMIP5 models basically reproduce the lagged or leaded relationship between the SST

anomalies and the LHF anomalies. When the SST anomalies lag the LHF anomalies, most models except for CanESM2,

CESM1-BGC and NorESM1-ME simulate a large region of positive covariance in the subpolar NA, which only occurs

in a small region in the observation-based results. When the two variables in the models are synchronized, the range and

magnitude of positive covariance simulated by models are significantly larger than those in the observation-based results.

It can be concluded that the oceanic forcing on the atmosphere through the LHF variation is enhanced in the models,

which results in the positive response of the winter-averaged SST anomalies to the LHF anomalies in the NA, which

weaken the magnitude of the negative response center of the SST anomalies to the NAO, and may be the reason of the

bias of the response of the SST anomalies to the NAO in the subpolar NA by NorESM1-ME which has a realistic

relationship of the SHF and SST. .

## 4    Conclusion

We evaluated the influence mechanism of the NAO on the SST in the NA (0-65°N) simulated by the 13 models of

CMIP5. In most of the models, the significant periods of interannual signals obtained by the power spectra are consistent

with the observation-based results, and the significant periods of the subpolar and subtropical area-averaged SST in the

observation are mainly characterized by interannual signals, so we mainly evaluated the simulation of the relationship

between the winter-averaged SST and NAO by these 13 CMIP5 models on the interannual scale.

Based on the observations, the RCs of winter-averaged SST anomalies against the NAO show a significant tripolar

distribution along the meridional direction in the NA. Most of the models can reproduce the tripole pattern of the response

of the SST anomalies to the NAO. In the subtropical NA (25-45°N), most models can reproduce the significant positive

response center near the American coast. However, in the subpolar region, the simulated locations and magnitude of the

negative response centers by most models have some difference from the observation.

Further evaluation of the response of the winter-averaged SST anomalies to the NAO simulated by the 13 CMIP5

models in the NA shows that the models can basically reproduce the impact of the wind speed anomalies related to the

NAO on turbulent heat flux anomalies in the NA, but the relationship between the anomalies of the NAO-driven turbulent

heat flux and SST simulated by the models has some differences from observation-based results in some regions of the

NA, especially in the region north of 45°N. In the subpolar NA, the geographical range and magnitude of the negative

response of the SST anomalies to the NAO-driven SHF / LHF simulated by most of the models are smaller than those

in the observation-based results, which may lead to the bias of the locations and magnitude of the negative response

centers of the SST anomalies to the NAO. One piece of evidence for this conclusion is that the bias of locations of

negative response centers of the SST anomalies to the NAO-driven SHF / LHF simulate by most models is corresponding

to the bias of locations of negative response centers of the SST anomalies to the NAO. It seems that the influence of the

LHF on the SST is weaker than that of the SHF on the SST in the most regions from both the observation-based and simulated results by most models, except for the eastern tropical NA and NorESM1-ME. The weak negative response or strong positive response of the SST anomalies to the NAO-driven LHF simulated by most models may be caused by the

rapid response of the ocean to the change of the LHF.

## 5    Discussion

Although the response of the SST anomalies to the NAO in most models can be explained by the bias of the SST response to the air-sea turbulent heat flux, there are still some models whose performance has not been reasonably explained, for example, MRI-ESM1 in which the relationship of the SST and the NAO-driven heat flux is not consistent

with the observation-based result in the high latitude but the response of the SST to the NAO is realistic. There may be some other factors which can affect the relationship of the SST and NAO, or deficiencies in the method used in this study.

### 5.1 Heat advection transport

In addition to the turbulent heat flux, the changes of long / short wave radiation and the ocean circulation also have

effects on the change of the SST. The long-wave radiation on the sea surface is mainly determined by SST, while the change of short-wave radiation does not have a strong relationship with the NAO (the figure is omitted). The simulated relationship between the SST and the NAO by the CMIP5 models may be also related to the NAO-driven horizontal heat advection, although some other studies have argued that the impact of ocean heat advection on the change of SST in the

subtropical NA is mainly on the decadal scale (Delworth et al., 1998, Krahmann et al., 2001).

From the observation-based RCs of surface seawater meridional velocity (vo) anomalies against the NAO in winter

(Fig. 10a), it can be seen that on the interannual scale, in the subtropical NA the observed vo (positive values indicate

northward) anomalies has a significant positive response to the NAO, and in the subpolar NA it has a significant negative

response to the NAO. The observed and simulated correlation coefficients between the winter-averaged SST anomalies

and the SST variations caused by NAO-driven vo on the interannual scale which are calculated with Eqs. 7-8 are shown

in Figure 10b. There seems no obvious regular distribution of the influence of the NAO-driven vo on the SST which can

be related to the tripole pattern of the response of the SST anomalies to the NAO (Fig. 10b), so on the interannual time

scale the role of the heat advection to the change of the SST in the NA can be ignored.

## 5.2 Band-pass filter

Cane et al., (2017) point out that the low-pass filter can influence the correlations of the SST and heat flux. Therefore,

the influence of the band-pass filter should be analyzed. We also did regression analysis of unfiltered winter average SST

anomalies and NAO indices (Fig. S7). It is found that except for the models of IPSL-CM5A-MR and MPI-ESM-L / MR,

there is no obvious difference in the distribution of standardized RCs of the SST and NAO between the filtered and

unfiltered results, and the main difference is that the RCs from the unfiltered data are slightly smaller than those from

the filtered data in the subtropical NA (Fig. 4) of both the observation-based results and most of the modeled results. In

IPSL-CM5A-MR and MPI-ESM-L / MR, both the magnitude and location of the positive significant RCs of the

unfiltered SST and NAO indices in the subtropical NA are changed and are much closer to the observation-based results than those of the filtered results. It indicates that in these three models the signals over 8 years probably play a more important role in the response of the SST to the NAO. It should be noted that in the tropical and subpolar NA the RCs of the SST against the NAO in the unfiltered observation-based results are enhanced, but those in most of the unfiltered modeled results are weakened. The area of the negative response of the SST to the NAO is enlarged in the unfiltered results of NorESM1-ME, and much closer to the observation-based results than that in the filtered results. The periods of the NAO and SST illustrated in Fig.3 do not provide us with enough information to explain the phenomenon occurring in the observation-based results and simulated results of NorESM1-ME, but they do indicate that the data filtering process can affect our evaluation of individual models to some extent. For the relationship between the SST and NAO-driven SHF / LHF (Fig. S8), the magnitude of the RCs of the SST and NAO-driven SHF / LHF anomalies is enhanced in the unfiltered observation-based results but weakened in most of the unfiltered model results except for the subtropical NA of IPSL-CM5A-MR and MPI-ESM-L/MR and the subpolar NA of NorESM1-ME. The difference between the unfiltered and filtered RCs of the SST and NAO-driven SHF / LHF anomalies in the models is consistent with that between the unfiltered and filtered $RC_S$ of the SST and NAO. Once again, the NAO-driven heat flux anomalies are the key to control the response of the SST to the NAO.

The periods of NAO indices are sensitive to the time period analyzed. The significant periods of the observed NAO index in 1897-2005 are 2.3–2.7, 4.7–5.8, and 8.3 years, but in 1955-2005 they become 3, 4.8, and 8-10. The periods of NAO indices are also sensitive to the dataset, which is analyzed in Jing et al., (2019). Therefore, the influence of the

cutoff period used in the filter should be analyzed. We do regression analysis of winter average SST anomalies and NAO

indices on the interannual scale calculated by 2-4 years filtering (Fig. S9) and find that the observed and simulated

patterns of the response of the SST anomalies to the NAO based on 2-4 years filtering is close to the results based on 2-

6 years but the intensity of the response of the SST anomalies to the NAO in the observation and most models is

strengthened. It should be noted that with the 2-4 year filtering the performance of the response of the SST to the NAO

in MPI-ESM-L / MR and NorES1-ME is very close to that with the 2-6 years filtering. Combined the difference between

the unfiltered and filtered results, it can be concluded that there are indeed signals on a long timescale which play more

important influence on the relationship of the SST and NAO, although we cannot get supporting information from the

analysis of periodicity.

## 5.3 NAO index definition

Currently, there are many NAO index definitions, because the method used to define NAO index differs, the

description of several associated phenomena will also vary (Pokorná and Huth, 2015). In addition to the site-based NAO

index provided by NCAR in this study, the two other observation-based NAO indices defined by Gong and Wang

(2000)'s method and the method used to calculate model NAO indices (Zheng et al., 2013) are applied to study the

relationship between different NAO indices and SST on the interannual scale (Fig. S10). The NAO index defined by

different methods does affect the relationship between the NAO and SST in the tropical NA, but does little effects on the

relationship in both the subpolar and subtropical NA. This suggests that if one focuses on the relationship of the SST

and NAO in the tropical NA, one should be careful of his choice of the NAO indices.

## 5.4 Initial fields and external forcing for models

Kay et al. (2015) did ensemble experiments by adding different minute perturbations to the atmosphere as initial conditions to study the internal variability. There are also some ensemble historical experiments in CMIP5 which are initialized with different initial conditions in 1850. The initial conditions of these ensemble members are from the different integration times of the piControl experiments, so these initial conditions represent the different time histories of internal variability. The relationship of the NAO and SST simulated by the models with differing initial fields (r1i1p1 and r3i1p1), as previously discussed, are compared (Fig. S11). There are 7 of 13 models which employed the historical experiment results with different initial fields. The locations of the response centers of the SST anomalies to the NAO simulated by 6 of the 7 models in the r3i1p1 experiments are close to those from the r1i1p1 experiments (Fig. S11), while the magnitude of the response centers simulated in the r3i1p1 experiments is stronger than that in the r1i1p1 experiments (Fig. 4). The locations and magnitude of response centers of the SST anomalies to the NAO simulated by MPI-ESM-MR in the r3i1p1 experiment are obviously different from those in the r1i1p1 experiment with the same filtering cutoffs, and are closer to the observation and the 2-4 years filtering results of the r1i1p1 experiment. The significant periods of the NAO in the experiments of r3i1p1 are also different from those of r1i1p1 (Fig. S12), but there is no obvious law about the significant periods and the magnitude of the response of the SST on the NAO in these 7 models. We cannot figure out the reason for the difference between these two sets of experiments, especially in MPI-

ESM-MR, but it should be emphasized that the influence of the initial conditions on the result needs to be considered in the evaluation of some individual models.

Besides the initial fields, the external forcing may impact the relationship of the SST and the NAO in models. To investigate the influence of the external-forcing data of the historical experiments, the outputs from the piControl experiments are used to compare with those from the historical experiments. The relationships of the NAO and SST from the piControl experiments of the 13 CMIP5 models are analyzed (Fig. S13). The locations of the response centers of the SST to the NAO are close to those in historical experiments of most models (except for the subtropical NA of

CESM1-BGC). The main difference between the piControl and historical experiments is the magnitude of the response of the SST on the NAO, which also occurs in the difference between the two sets of historical experiments (r1i1pi1 and r3i1p1). It should be noted that the result of the piControl experiments in MPI-ESM-MR is very similar with the historical experiments (r3i1p1), but is different from the historical experiments (r1i1p1). The response of the SHF/LHF to the NAO in the piControl experiments (Fig. S14) is also stronger than that in the historical experiment (r1i1p1, Fig. 7). Based on

the above analysis, it can be concluded that the influence of the external forcing has less influence on the NAO-SST relationship in most models, especially on the locations of response centers, but in some individual models the influence of external forcing cannot be ignored. Some studies have shown that in the climate models, the amplitude of the response to the external forcing (such as volcanic forcing, solar variability, and ozone depletion) is weak, which leads to weak predictable signals in these models although these models can predict observed climate variability (Scaife and Smith et

al. 2018). The weak predictable signals inhibit the estimation of forced climate variability in the Atlantic sector (Scaife

and Smith et al. 2018). The weak influence of the external forcing on NAO-SST relationship was also found in the

CMIP5 models in this work. Scaife et al. (2020) have argued that a large ==ensemble number== can overcome the signal-to-

noise paradox, which probably provide a reference for the future application of CMIP models in the predications.

Data availability.

The addresses of downloading all data used in this study have been described in Sect. 2.1. All data for results are available by contacting the corresponding author.

Author contributions.

YJJ, YCL and YFX designed the research. YJJ analyzed the data under the guidance of YCL and YFX, and prepared the manuscript with contributions from YCL and YFX.

Competing interests.

The authors declare that they have no conflict of interest.

Acknowledgements.

All the authors thank the European Centre for Medium Range Weather Forecasts (ECMWF) for providing the sea level pressure data, the Hadley Centre Global Sea Ice and Sea Surface Temperature (HadiSST) for providing sea surface temperature data, and National Oceanic and the Atmospheric Administration (NOAA) for providing the 10-m wind speed, turbulent heat flux and sea water meridional velocity data. The authors thank those institutions that developed the CMIP5 Earth System Models and provided simulation results. Finally, the authors also thanks the reviewers and topic editor for their conscientious reviews and useful comments on an earlier versions of this paper.

Financial support.

This work was supported jointly by the National Key Research and Development Program of China (No. 2016YFB0200800), the National Natural Science Foundation of China (Grant No. 41530426) and the Strategic Priority Research Program of the Chinese Academy of Sciences (Grant No. XDB42000000).

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

Table 1. CMIP5 models used in this study.

| models | Sponsor, Country | Ocean Model | Marine Resolution (lat×lon) |
|---|---|---|---|
| CanESM2 | Canada, CCCMA | CanOM4 | 0.98º × 1.4º |
| CESM1-BGC | U.S.A, NSF-DOE-NCAR | POP2 | 320 × 384 grid points (gx1v3) |
| GFDL-ESM2G | U.S.A, NOAA-GFDL | GOLD | 0.6º × 1.0º (tripolar) |
| GFDL-ESM2M | U.S.A, NOAA-GFDL | MOM4.l | 0.6º× 1.0º (tripolar) |
| HadGEM2-CC | Britain, MOHC | HadGOM2 | 0.3º–1º× 1º |
| HadGEM2-ES | Britain. MOHC | HadGOM2 | 0.3º–1º× 1º |
| IPSL-CM5A-LR | France, ISPL | ORCA2 | 2º× 2º |
| IPSL-CM5A-MR | France, ISPL | ORCA2 | 2º× 2º |
| IPSL-CM5B-LR | France, ISPL | ORCA2 | 2º× 2º |
| MPI-ESM-LR | Germany, MPI-M | MPI-OM | 1.5º×1.5º |
| MPI-ESM-MR | Germany, MPI-M | MPI-OM | 0.4º× 0.4º |
| MRI-ESM1 | Japan, MRI | MRI-COM3 | 0.5º×1º |
| NorESM1-ME | Norway, NCC | MICOM | 0.5º×1º |

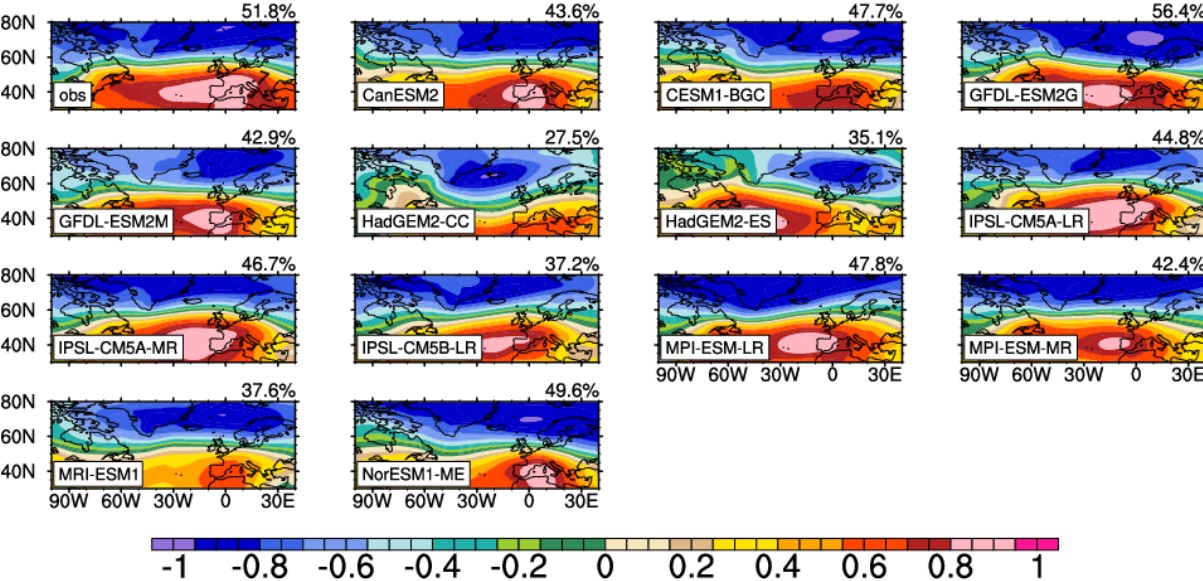

Figure 1 EOF1 of observed and simulated standardized winter-averaged sea level pressure over the particular region of the North Atlantic (30 -80°N; 100°W-40°E). The time periods for the observation and models range from 1965 to 2015 and 1955 to 2005, respectively. The simulated results are based on historical experiment of CMIP5 (r1i1p1).


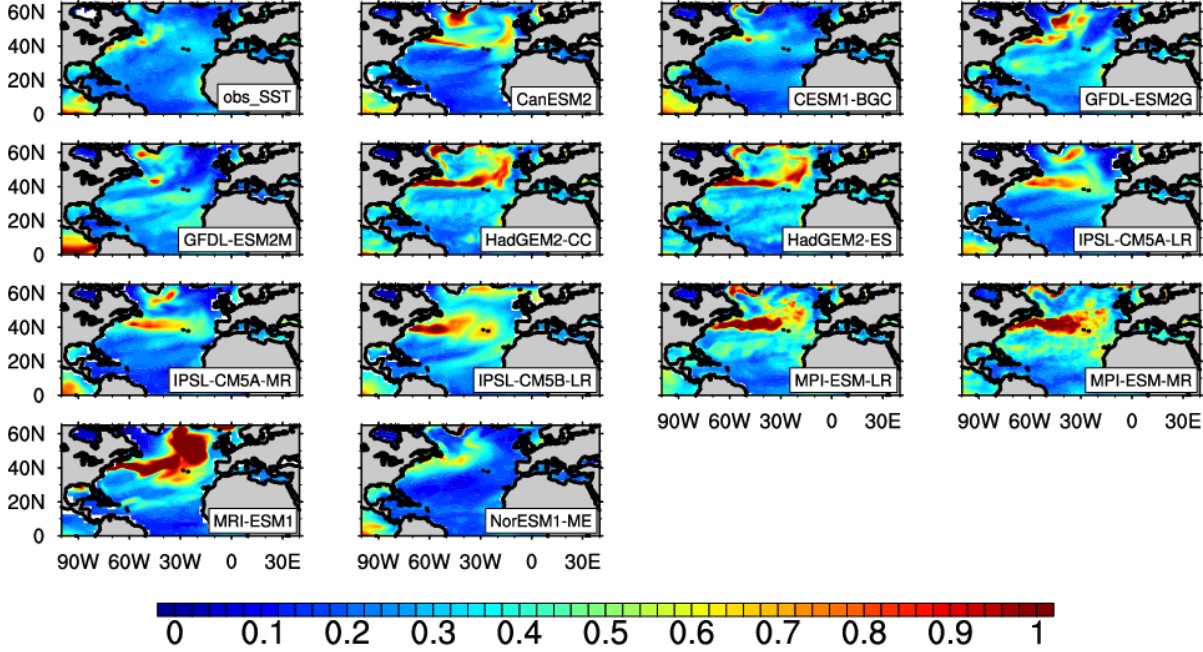

Figure 2 Observed and simulated SST interannual variability (℃, 1SD). The time periods for the observation and models range from 1965 to 2015 and 1955 to 2005, respectively. The simulated results are based on historical experiment of CMIP5 (r1i1p1).

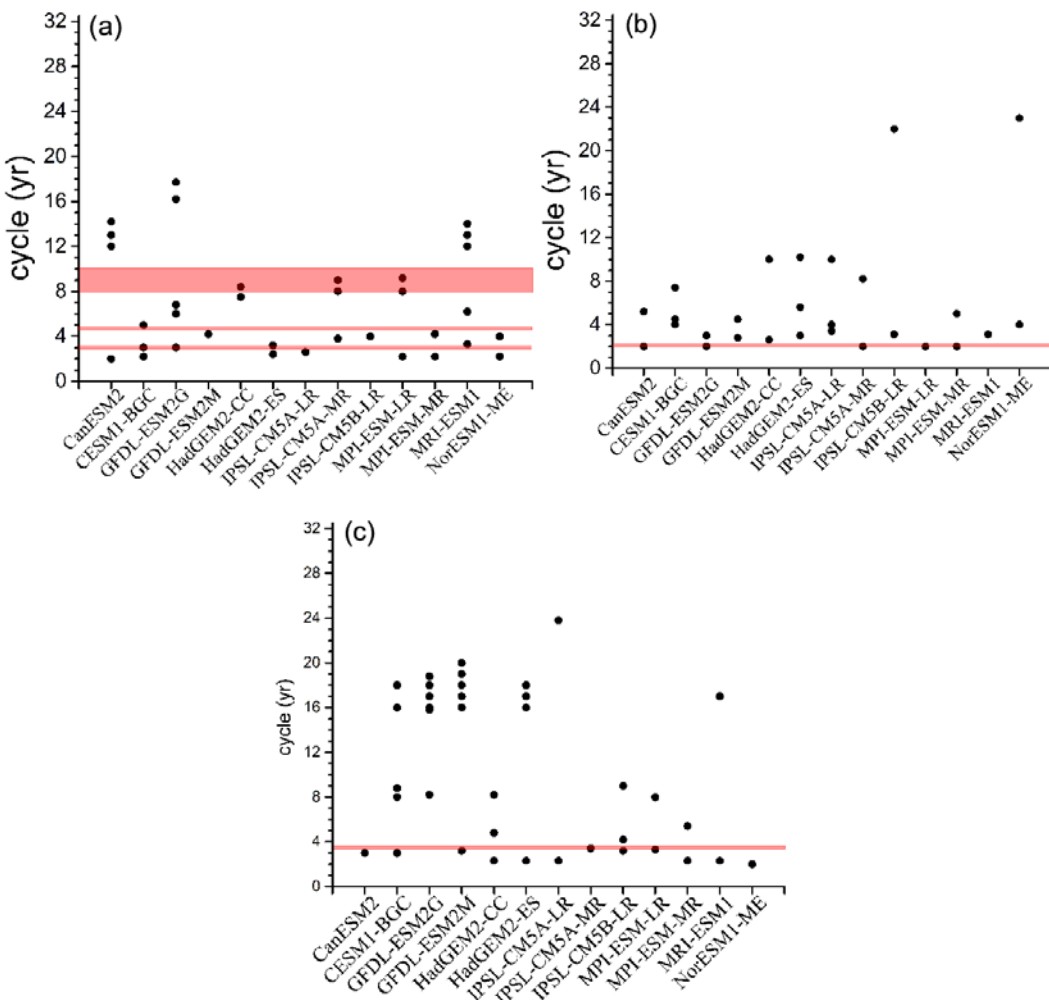


Figure 3 Periodicities of the observed and simulated winter-averaged NAO indices (a), area-averaged SST anomalies in the subtropical (b, 25-45°N) and subpolar NA (c, 45-65°N), determined by power spectrum analysis. The periodicities are determined by calculating the red noise confidence interval and choosing those at the 90% confidence level. The Y coordinate of the horizontal lines / areas is the significant period of observation. The time periods for the observation

and models range from 1965 to 2015 and 1955 to 2005, respectively. The simulated results are based on historical experiment of CMIP5 (r1i1p1).

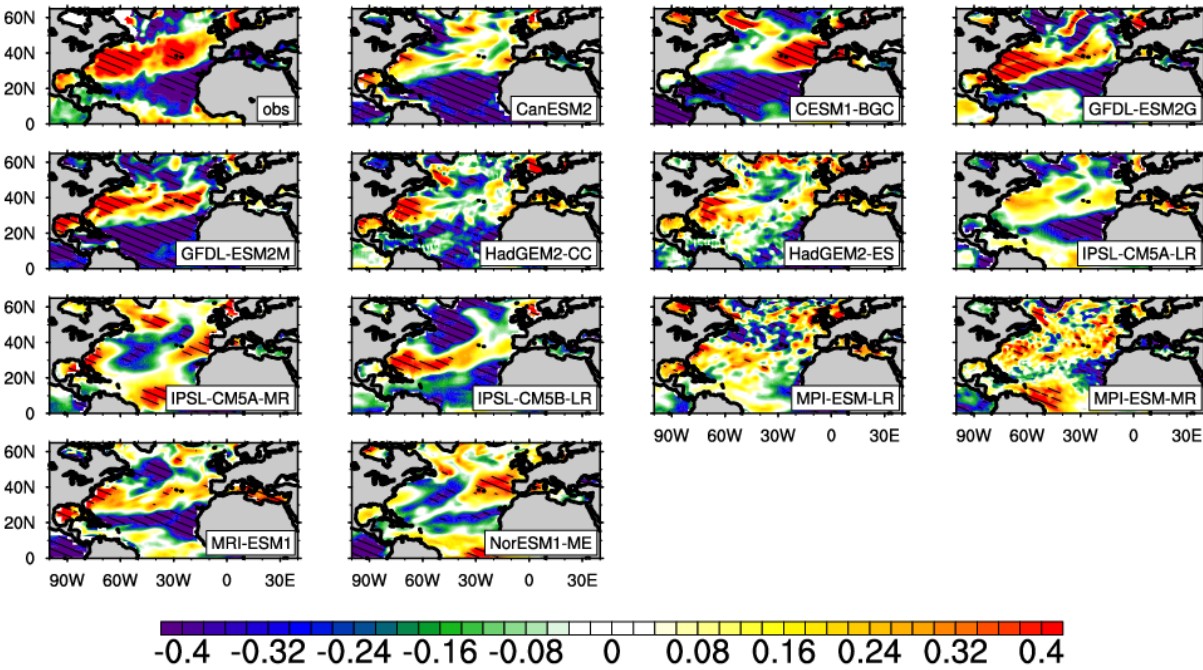

Figure 4 Observed and simulated standardized regression coefficients (RCs) of the winter-averaged SST anomalies against the NAO indices on the interannual scale (with 2-6 years data filtering). Shaded areas indicate that RCs are statistically significant at the 95% confidence level of the Student's t-test. The obs is the RCs of observed SST to the NAO indices provided by NCAR. The time periods for the observation and models range from 1965 to 2015 and 1955 to 2005, respectively. The simulated results are based on historical experiment of CMIP5 (r1i1p1).

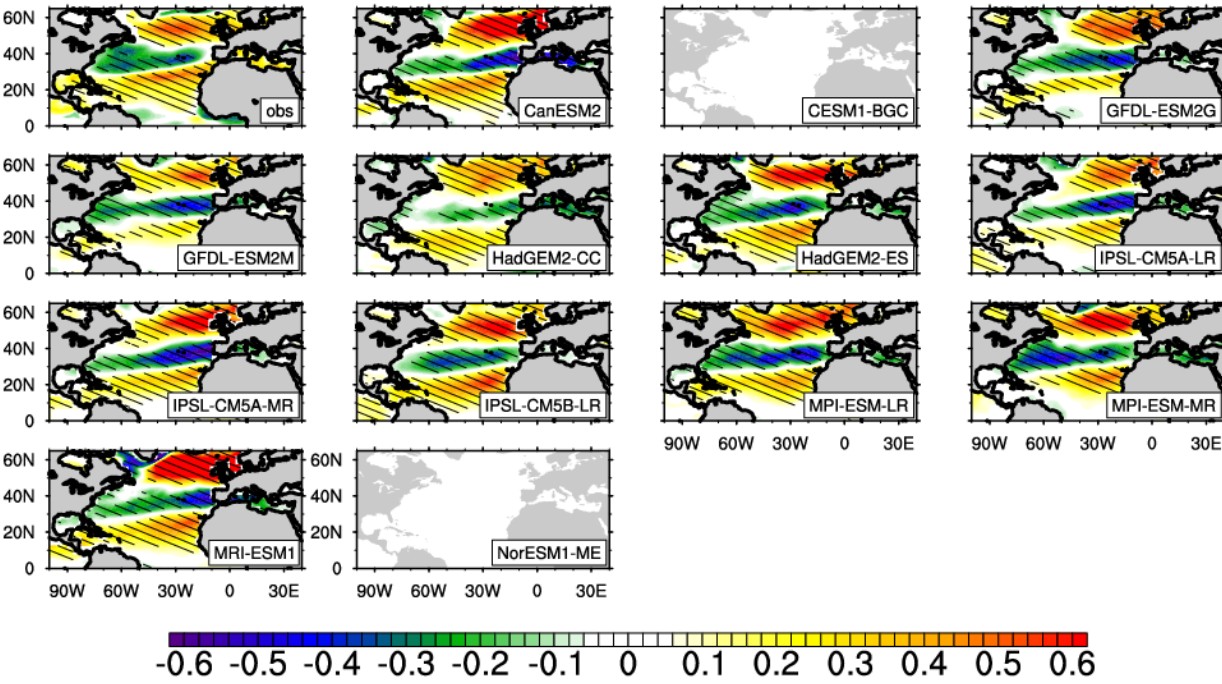

Figure 5 Same as Fig.4 but for the winter-averaged sea surface wind speed anomalies against the NAO indices. A missing panel means that the model output is not available.

(a)

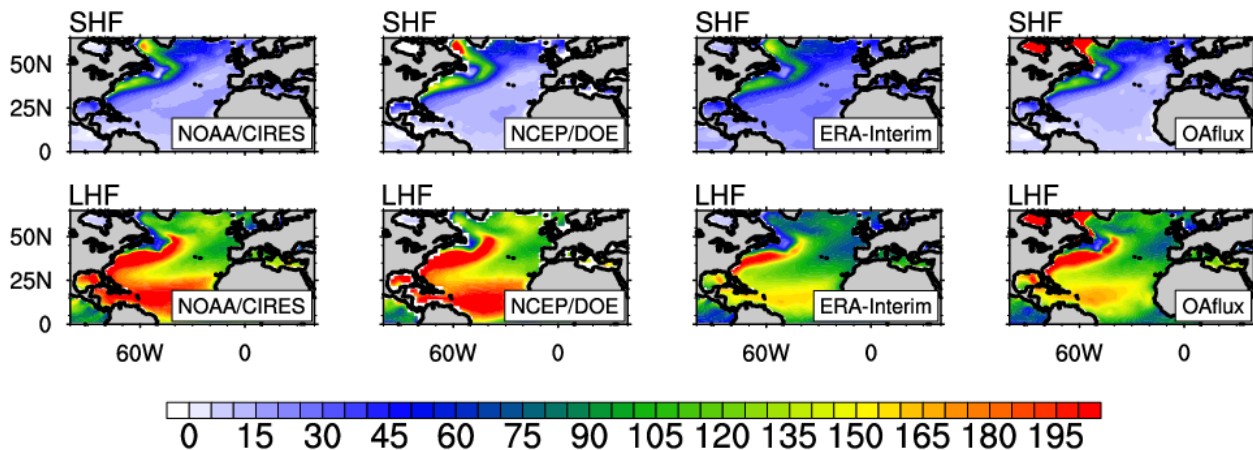

(b)

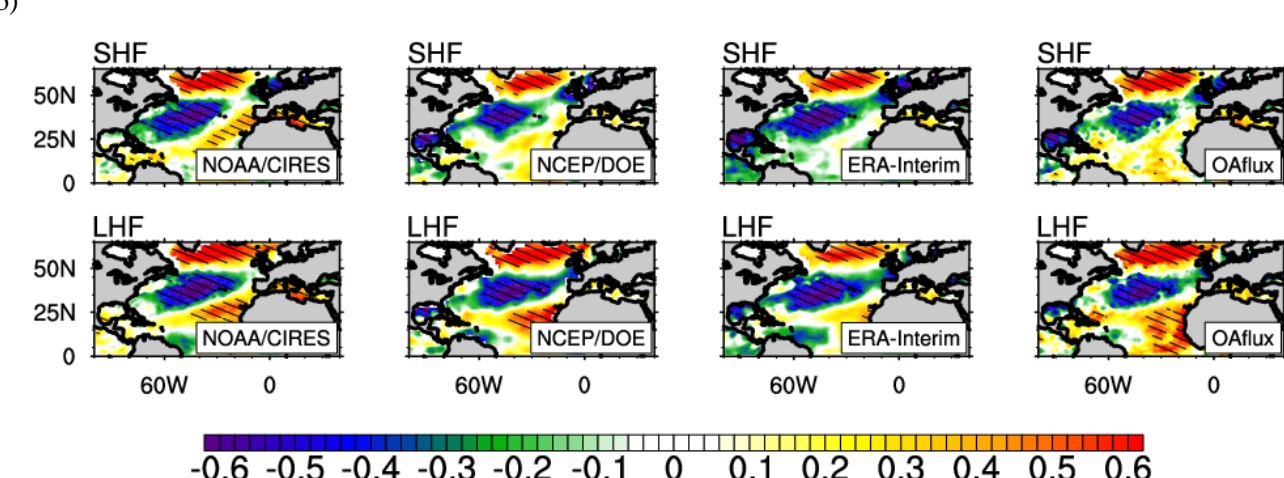


Figure 6 (a) Observed multi-year mean winter sensible (SHF) / latent heat flux (LHF) (Wm$^{-2}$), and (b) standardized RCs of the winter-averaged SHF/ LHF anomalies against the NAO indices. The time periods of the 4 datasets used in this figure range from 1980 to 2015. The three observation-based data from the NOAA-CIRES (NOAA/CIRES), NCEP (NCEP/DOE) and ECWMF (ERA-Interim) are reanalysis data, and the last one (OAflux) is the optimal

syntheses of voluntary observing ships, reanalysis, and satellite data sources.

(a)

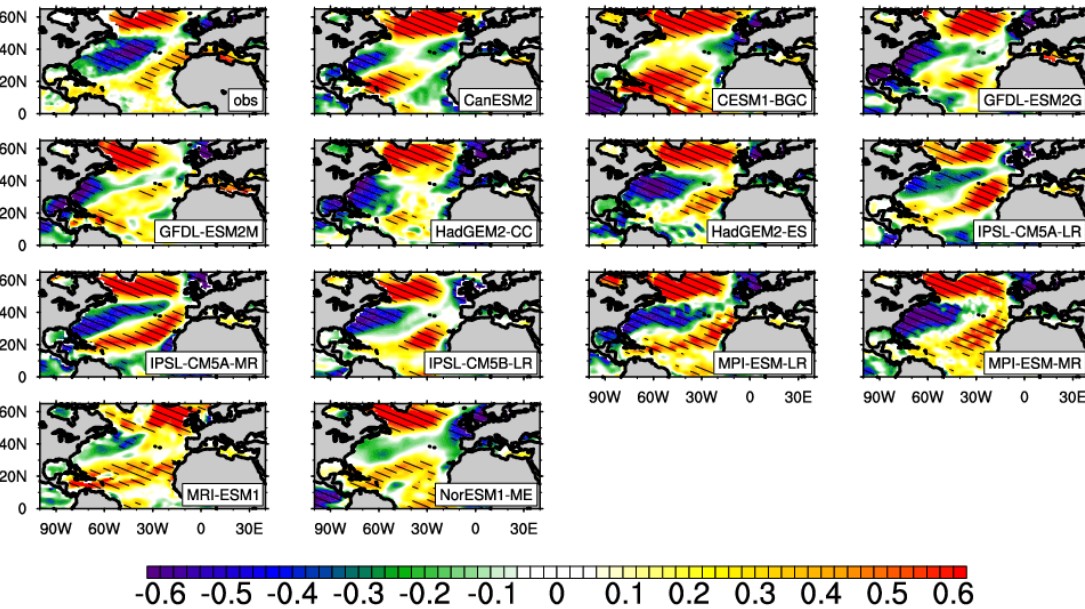

(b)

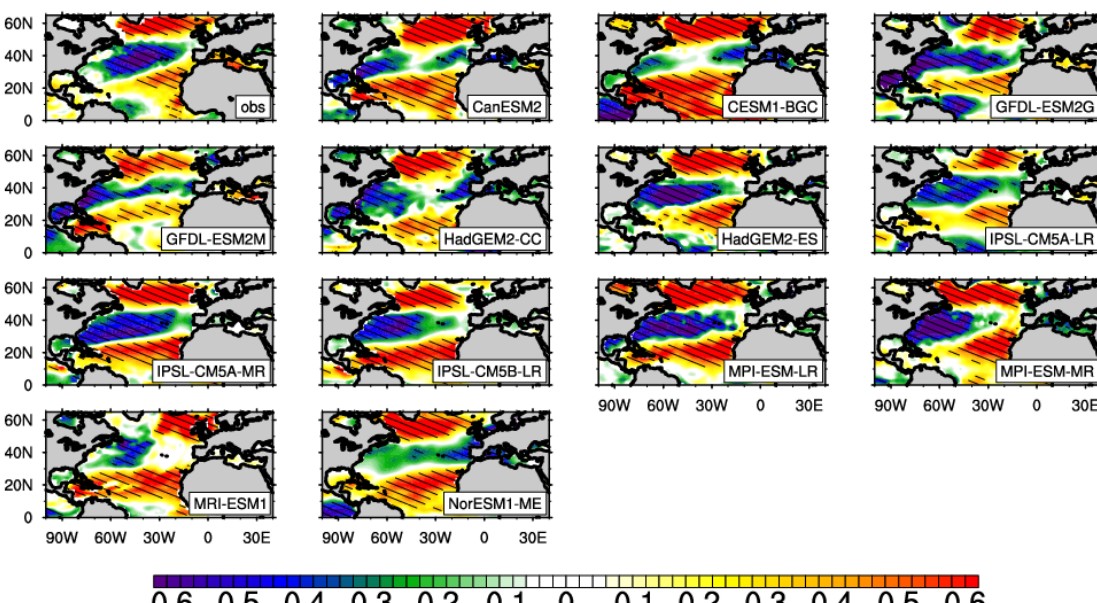

Figure 7 Same as Fig. 4 but for the winter-averaged SHF (a) / LHF(b) anomalies against the NAO indices.

(a)

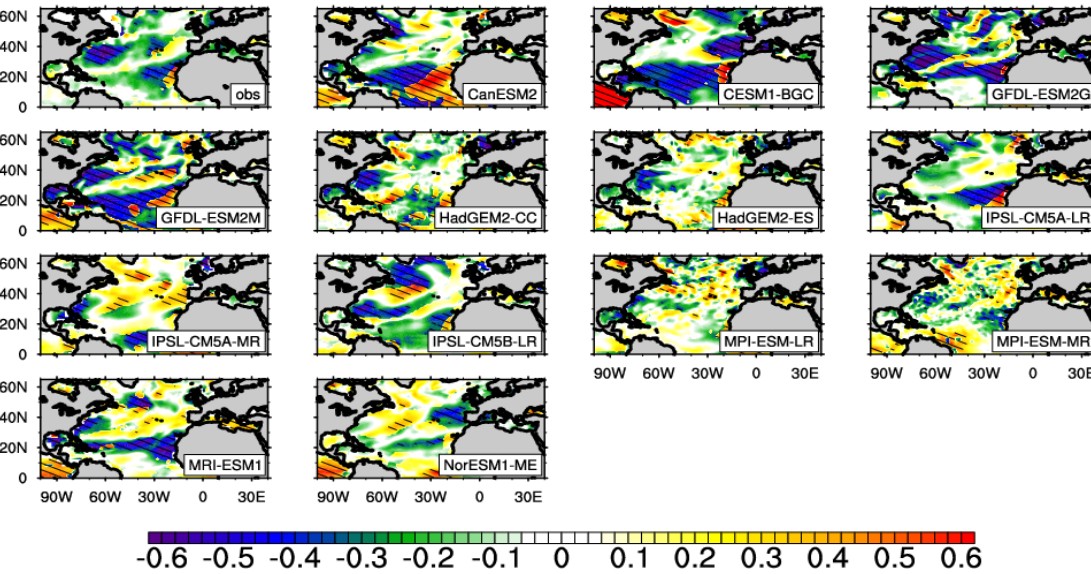


(b)

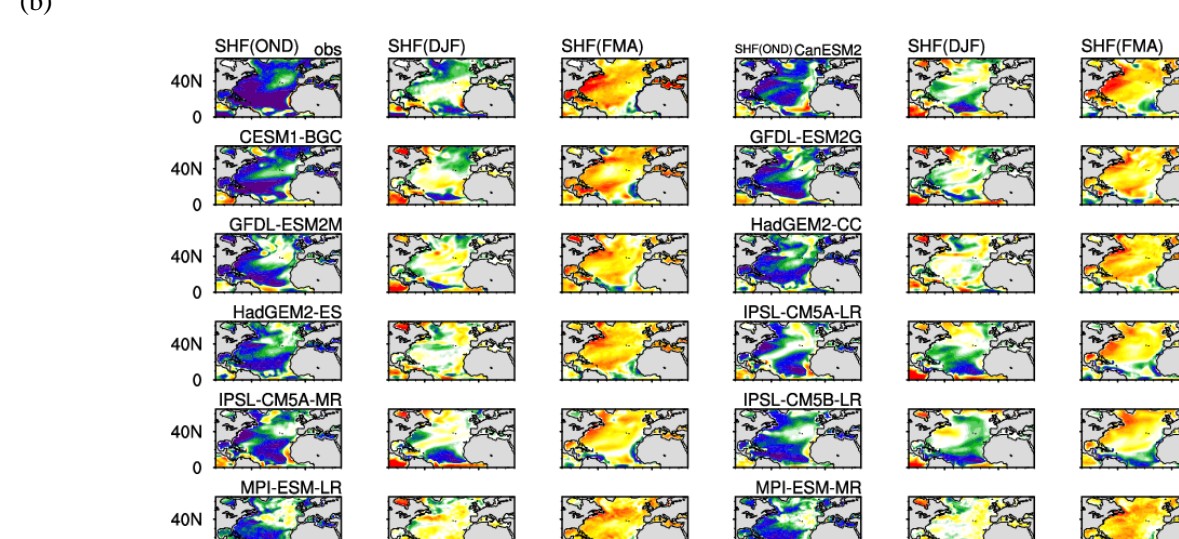

Figure 8 (a) Same as Fig. 4 but for the winter-averaged SST anomalies against the NAO-driven SHF anomalies. (b)

The standardized lagged (leaded) covariance between monthly SHF anomalies from October to December / from February to April with the SST anomalies from December to January. OND refers to the SHF from October to December, and FMA refers to the SHF from February to April.

(a)

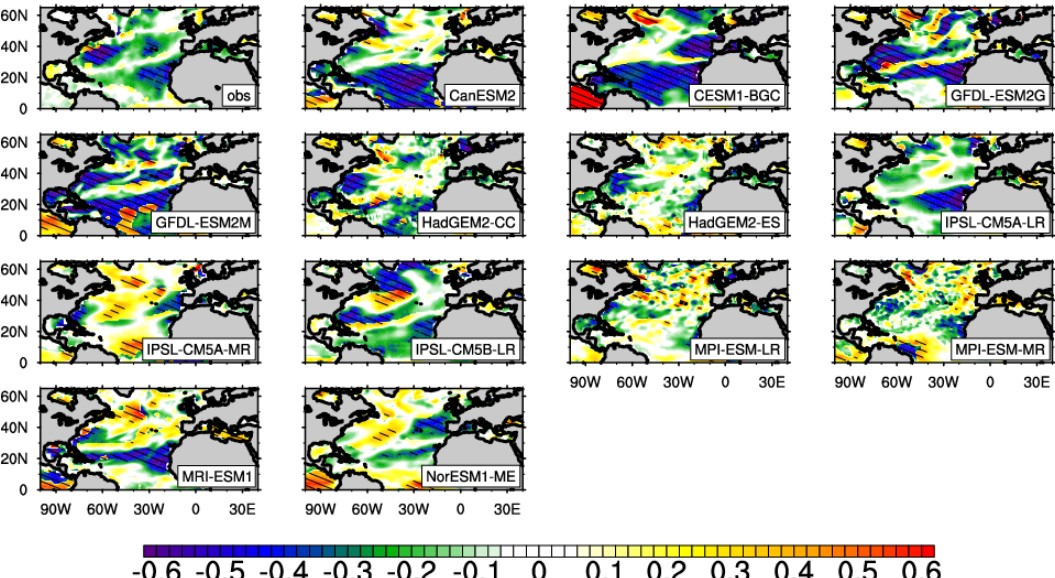

(b)

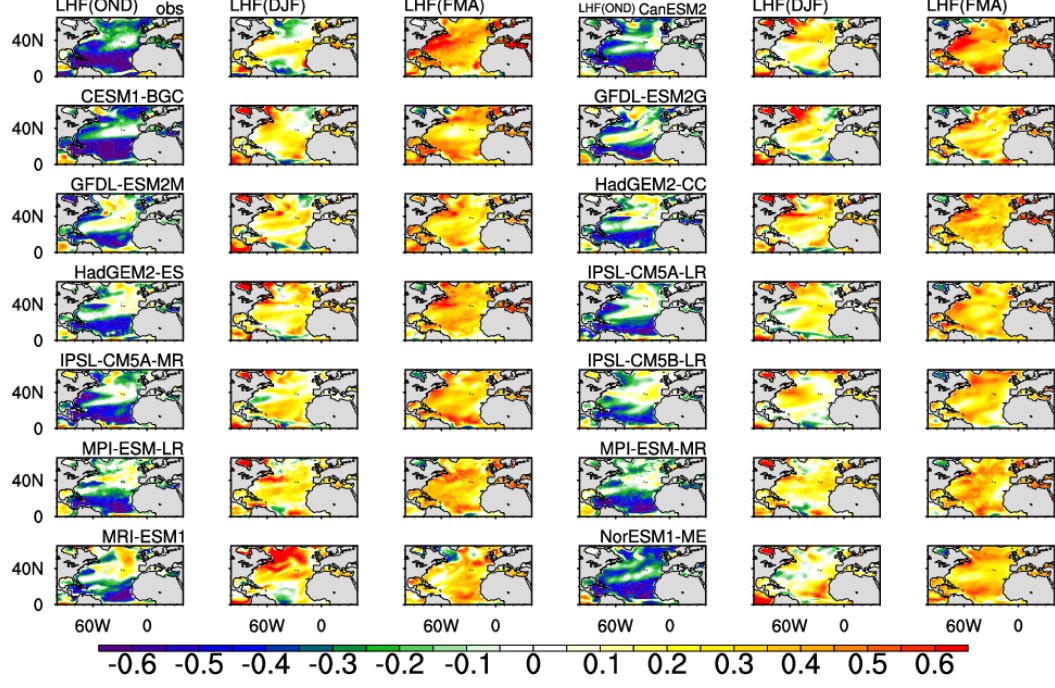

Figure 9 Same as Fig. 8 but for the LHF

(a)

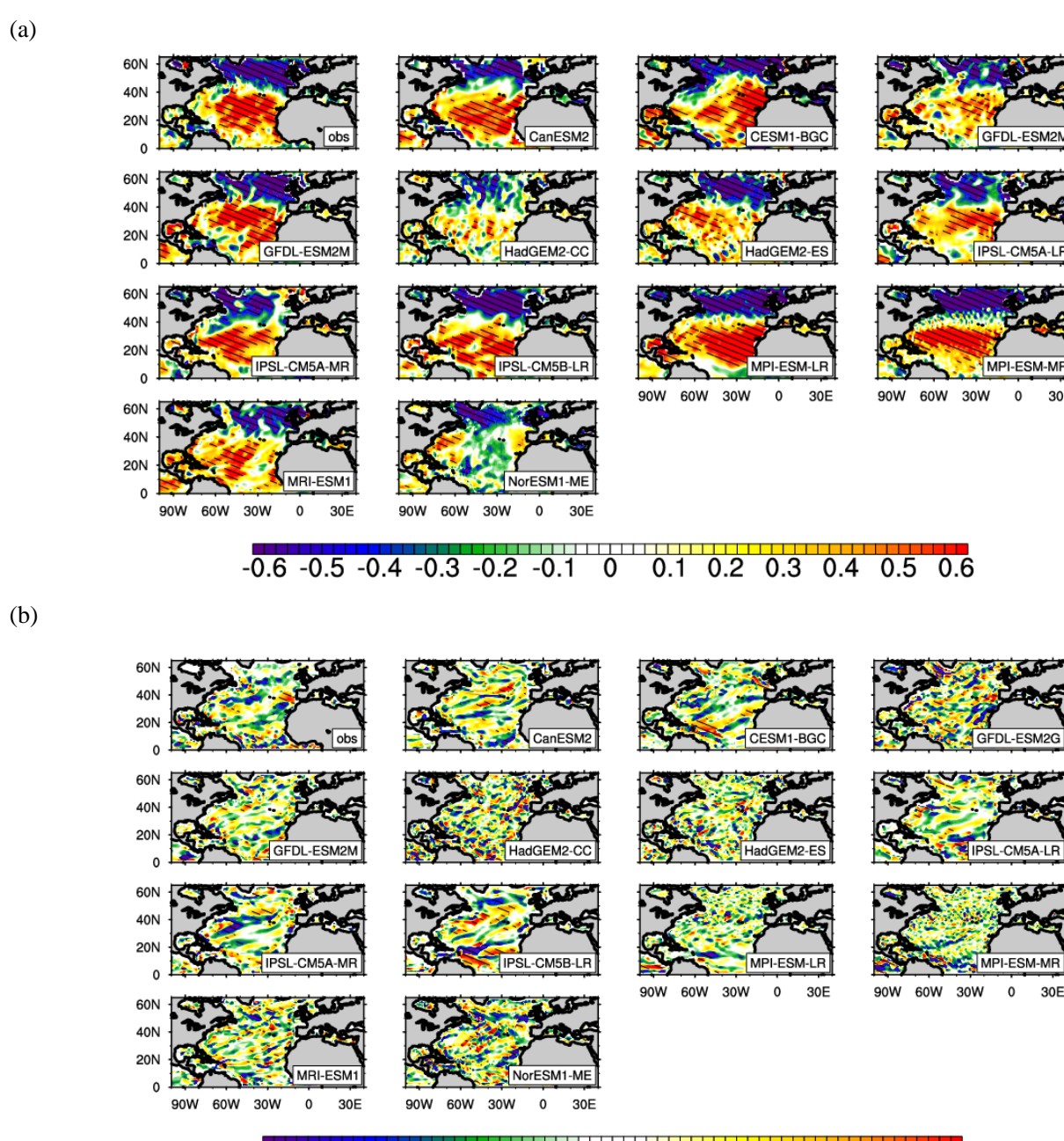

(b)

Figure 10 (a) Observed and simulated standardized RCs of the winter-averaged sea surface meridional velocity anomalies against the NAO index, (b) Standardized correlation coefficients between the winter-averaged SST

anomalies and the change in SST caused by NAO-driven surface meridional velocity on the interannual scale (with 2-6
year data filtering, equations 7-9). Shaded areas indicate that the correlation coefficients are statistically significant at
the 95% confidence level of the Student's t-test. The obs is the observation-based result. The time periods for the
observation and models range from 1981 to 2015 and 1955 to 2005, respectively. The simulated results are based on
historical experiment of CMIP5 (r1i1p1).
