# Peer review of "Assessment of responses of North Atlantic winter SST to the NAO on the interannual scale in 13 CMIP5 models"

_Ocean Science, 2020_

## Referee Comment (RC1) · Anonymous Referee #1 · 11 Jun 2020

General comments:

The manuscript addresses the response of winter SST in the North Atlantic to NAO forcing in 13 CMIP5 models. Patterns of observation and models are compared. Because the patterns of NAO-driven SSTs look different between models and observation, the manuscript further investigates the link between the surface heat fluxes and the SST in the North Atlantic. Also, the link between the meridional ocean velocity and the NAO is presented. The authors suggest that it is the overestimated role of the ocean that causes an unrealistic relation between heat fluxes and SST, that finally cause the model to simulate a NAO-driven SST response that differs from the observed

pattern.

The basic motivation for the authors of this study is that the observed tripolar SST response to the NAO is simulated only by 7 models (as they write). However, I would say that more than only 7 model reproduce a tripolar pattern associated with the NAO on this timescale, even though the centres may be partly displaced, or not providing the correct amplitude (Fig. 4). I agree that there are models that cannot reproduce a realistic pattern, but most of them simulate a tripolar pattern. Especially, the positive centre near the American coast, is reproduced by all models. So, for me the research question would rather be, why are the subpolar (negative) centres displaced.

I would hypothesize, that the displacement of the location could be explained by: the wrong location of the NAO-driven heat flux forcing, a different mean ocean circulation in the models, a different response of the circulation to the NAO, or a combination of these aspects. The authors suggest that the cause for the unrealistic SST response is the incorrect response of SST to heat flux forcing, or as they write in the abstract for the subpolar North Atlantic 'most of the models simulate a positive response of SST to the turbulent heat flux'. And here I see a fundamental problem: When positive flux anomalies (ocean to atmosphere) are correlated with positive SST anomalies, then the SST is the driver for this link, not vice versa. Therefore, the regression that their conclusions are based on (Fig. 8 / 9), do simply not reflect the 'response of SST to the NAO via heat flux forcing'. The regressions seem, instead, to pick up something else, which may or may not be indirectly related to the NAO (for example through an ocean feedback). It could be that for some models / regions on the analysed timescales the dominant link between SST and heat fluxes is not the NAO-heat fluxes forcing the SST. To really extract the response of the SST to NAO-driven heat flux anomalies, I would compute regressions of SST on a pattern of the heat fluxes that has been shown to be is NAO-driven for each model (maybe an index representing the typical structure as seen in Fig. 7).

In summary, I see a fundamental problem with the interpretation of the results and

based on that also not enough evidence for the conclusions presented here.

Another issue is, that the motivation for this study are the differences in the SST-response to the NAO (Fig. 4). But already within the same SST dataset, there are differences depending on how the NAO index was calculated (the first two panels in the first row of Fig. 4). Next, if I understood correctly, the NAO index in the models is calculated with another (third) method. So, it can be assumed that a part of the differences is explained by how the NAO index was calculated.

Furthermore, the entire manuscript would need substantial improvements regarding grammar / language in general. Therefore, I didn't list all the language issues or unclear formulations, because there were just too many.

Based mainly on the concern that I have regarding the approach / interpretation of the results, I cannot recommend this manuscript for publication.

Specific comments:

I suggest to modify the title to not have 'CMIP5 models on the interannual scale' together. So maybe 'Assessment of responses of North Atlantic winter SST to the NAO on the interannual scale in 13 CMIP5 models'.

15: Please clarify on the word 'obvious', in observations or models?

20: For the sub-tropical region an 'incorrect positive response' is mentioned. Why is it 'incorrect' when this subtropical centre of the tripolar pattern should be positive? Further down it is also written 'models can simulate the realistic positive response of SST anomalies to the NAO in the subtropical NA'. which seems to be a contradiction.

Overall, I find the abstract hard to understand.

The timescales are not made clear in the abstract.

59/60: 'In recent years, more and more people have realized that the evaluation of the CMIP5 Earth System Models (CMIP5-ESMs) is the basis for study by these models.' I

am not sure what the authors are trying to say here?

64/65: 'unreasonable simulation of AMOC'. In which way unreasonable?

Why were these 13 models chosen? I would assume that SST and heat fluxes are widely available across CMIP5 models. Still, it seems that even out of this 13, two do not provide wind speed (Fig. 5 and 6).

What has been done with trends in the data, especially, when computing regressions?

Why do the 'regression coefficients' not have units? Are we actually looking at correlation coefficients here? What about the units for the covariances shown in Fig. 8b and 9b?

Is heat flux computed manually or is it a model output? And are the heat flux measures of observations and models derived /computed in a consistent way?

100: Please explained how the 'site-based' index is computed. Because it causes different regressions patterns (as seen in Fig. 4).

145: How exactly 'normalized'?

157: I don't understand this sentence: 'Because the locations of the NAO action centers simulated by most of the CMIP5 ESMs in different NAO phases do not show the movements illustrated by the observation (the figure is omitted), the differences between the models are not caused by the NAO period or the phase of the initial sign, but are only related to the structures of models.' So in observations the NAO pattern is not symmetric? And in the models it is symmetric? Or are the patterns also not symmetric, but differing from the observations? Also, I would argue that it is enough to say that - given the long period of 108 years and the rather short-timescale behaviour of the NAO - there is no reason to think the initial state would matter. But maybe I understood wrong what this sections was supposed to say.

179 / Fig. 3b: I do not see the value of analysing the power spectra of the SST averaged

over 0-65°N in the North Atlantic in this case. The NAO fingerprint on SST is tripolar, and even if not perfectly tripolar in the models, it is non-uniform. Thus, SST variability that is associated with the NAO when analysing this area-average, is at least partly averaged out. Such an index would rather yield he AMV influence. I suggest to use a different SST index or remove this panel.

185: 'Based on the above analysis, simulated periods of the NAO indexes and area-averaged SST anomalies on the decadal scale are different from the results of observation.' I don't see the data that clearly support this statement. It would be helpful to see the individual power spectra, instead of only the periods of the peaks.

187: 'mainly reflected [...] on the interannual scales'. But aren't Eden and Jung 2001 focusing on the inter-decadal scale of the SST response to the NAO and the role of ocean dynamics?

194: Six models are named which a have a positive response in the subpolar region. However, some of these have also an area with a negative response in the subpolar latitudes (besides the positive one). So, if being generous with the exact location of the subpolar centre, nearly all models (maybe except for IPSL-CM5A-MR and MPI-ESM-MR) show some kind of tripolar response to the NAO. Please comment on that.

199: This last sentence in this paragraph should be revised. Both GFDL models have a similarly strong positive centre as in the obs_Gong panel. But again, I really recommend using the same method to compute the NAO index.

215: Is the SHF and LHF computed or is it model output? This is not clear, because in 130 it is only write 'usually calculated' and equations are provided. Based on that, are the observational heat flux data obtained in the same way?

219 / Fig. S2: I cannot agree on the statement that all models overestimate the SHF north of 50°N. First of all, the observations do not cover the area in the Labrador Sea, which seems to be the area of maximum ocean heat loss in the observations, which

also seems to be the case for some of the models. So, I would say that, for example, both MPI models are doing quite well in reproducing the observed heat flux. Next, it is interesting that specifically the IPSL-CM5B-LR model is the one that is least 'overestimating' – I would rather say 'underestimating' the heat flux. And as mentioned before, I find this model least capable of reproducing the tripolar SST pattern associated with the NAO. In summary, I don't find it convincing that an overestimated heat flux (in the mean state) might be the cause for an unrealistic SST fingerprint in the models.

Another issue is, that I am not sure how robust the 'observations' regarding their heat flux mean states are (Fig. S2). I recommend to test that through showing only the heat fluxes during the last decades when higher quality and quantity of observations were available, and also showing the mean state of a different reanalysis product. It could be that they are indeed robust. It just needs to be shown, because the model performances are evaluated based on these results.

224: For SHF I would even say 30°-65°N. And also in the Gulf of Mexico and in the Caribbean. But again, it should be shown that these results based on the reanalysis product starting around the year 1900 are robust.

Fig. 6: I don't think Fig. 6 is useful. Naturally, increased wind speed tends to increase the heat flux (whether from ocean to atmosphere or atmosphere to ocean). When trying to explain the differences in the response of SST to the NAO, it would be more useful to compare the differences in quantities regressed onto the NAO index (like Figure 7), because as shown before (in Fig. 5), the wind-speed response to the NAO is non-uniform.

238: Please also comment on the comparison of models.

Fig. 8a: In the models with unrealistic positive correlations, is the atmospheric forcing (NAO) maybe too weak compared to other models / observations? It might be like that – when I compare the explained variances from Fig. 1.

310: 'NAO-driven SHF / LHF anomalies': Regression between SST and HF without a direct relation to NAO were shown. Therefore, it is not justified to say 'NAO-driven'. This could only be said if the regressions had been done on an index (e.g., PC-based) that is related to the NAO.

336-339: I cannot agree on this statement, because: Fig. 7 (for heat fluxes) is the analogous version to Fig. 10 (for the meridional ocean velocity). Both figures show that the models reproduce the observations. Based on that only it is not justified to say that the root for an unrealistic SST response to the NAO are the heat fluxes. Indeed, the regressions of heat fluxes and SST in Fig. 8 and 9 show that the heat flux / SST relations are not realistic. But an analogous analysis for the meridional surface ocean velocity / SST is not presented. Even here there might be differences. And then one cannot say that unrealistic aspects in the SST response are caused only by the wrong heat flux response.

348: 'Because there is a deviation between the simulated and observed periods of the NAO indexes / area averaged SST on the decadal scale , ...' What is meant by 'period' and 'deviation' here exactly? The most dominant timescales of variability from the power spectra?

363: 'LHF and SST is mainly related.' What is meant by 'mainly related'?

364: When the response to LHF is 'unreasonably positive', how can contribute to a too weak positive response in the subtropical NA (as mentioned further above in l. 355)?

366-367: 'have a significant positive response to the NAO'. This only applies to the subtropics, not the subpolar region.

As a supplementary document already exists, I would suggest to also show the individual power spectra from which Figure 3 is derived. For the power spectra please also provide the information about the window that is used to compute them.

Technical corrections:

28: First sentence: A 'relationship' is not an 'event'.

220: By 'MPI-ESM1' you probably mean MRI-ESM1?

352: 'along the meridian' – which one?

355: 'weaker positive responses'. I would add 'than observed'

366: 'observed meridional velocities'. Please add the information that it is the surface ocean velocity.

'Constant field value is 10' is not a good annotation for a panel where model data are not available.

In the figures with subpanels for the different models, sometimes there is one, sometimes there are two observational panels. That moves the position of the model panels and it is hard to compare them across figures. I suggest to have the observations as the last panels, or leave the second panel position free for the case there is no second observation panel.

Please increase the resolution of Figs. 1 and 3.

I think the SCCs in Figure 2 are not a very representative measure, because they hardly vary despite the model differences (as seen on the maps, or in the RMSE).

Figure 3: Please explain the meaning of the horizontal lines / areas.

---

## Referee Comment (RC2) · Anonymous Referee #2 · 19 Jun 2020

General Comments

This manuscript shows that in several CMIP5 historical runs, the wintertime sea-surface temperature (SST) response to the North Atlantic Oscillation (NAO) is not consistent with observations, on interannual timescales. The authors demonstrate that some models that exhibit this discrepancy fail to reproduce the observed relationship between SST and turbulent heat fluxes (TFH), particularly in the subpolar gyre. They attribute this to an over-influential ocean. Most models examined here correctly produce the interannual NAO-SST relationship in the subtropical Atlantic because of the larger influence of Ekman forcing.

[Figure]

Overall, this manuscript asks a compelling question and applies a reasonable mechanistic approach to answering it. However, there are several questions I would like to see addressed before I can recommend publication (listed below). In addition to comments below, I think this work would be significantly more impactful with further copyediting.

1. Do the results show model dependence or sensitivity to initial conditions? I think readers will find the results to be more convincing if the authors could rule out dependence on initial conditions in the NAO – SST relationship. One way of doing this might be to look at a single-model large ensemble (e.g. CESM-LENS; Kay et al. 2015). For unfiltered output, it seems like the authors' results should hold (http://webext.cgd.ucar.edu/Multi-Case/CVDP_ex/cesm1.lens_1920-2018/nao.tempreg.djf.png) – but is that still true when bandpass filtered?

2. If the results show model dependence, how do the different models responses to historical forcing influence your results? On line 321, the authors note that the NAO has a negligible response to shortwave radiation – but shortwave radiation does affect SST. Can we be sure that the SHF and LHF are not responding to externally-forced changes in SST over this twentieth century, with the NAO being a bystander? In other words, if the models are responding in different ways to historical forcing, is that alone enough to change the relationship between the NAO and SST/SHF/LHF? One way to address this question may be to look at pre-industrial control runs of the same CMIP5 models the authors already examine.

3. Can the authors make causal claims based on band-pass filtered output? Cane et al. (2017) show that it is difficult to make causal claims about the sign of the relationship between heat fluxes and SST based solely on low-pass filtered data. I think that to show the causal relationship that is described in-text, it would be useful to also present unfiltered/annual average/wintertime average plots so that readers can be sure that the relationships shown are not an artifact of filtering. If the unfiltered results are similar, I think that is worth mentioning in-text.

[Figure]

Specific Comments

Lines 41 – 45: The wording and structure here is a bit too close to Deser et al. 2010 (page 119). Please edit.

Line 46 – 48: I would recommend re-wording to emphasize that the "uniform SST warming" occurs after the NAO+ in observations and pre-industrial control runs of climate models (see Delworth et al. 2017). Further, I would caution that this mechanism is not established as causal in observations (e.g. Buckley and Marshall 2016).

Line 64: Please clarify that Wang et al. (2014) found the NA SST was "underestimated" and "unreasonable" relative to observations (i.e. biased).

Lines 157 – 160: So, the NAO centers of action are stable with time in models, but not observations? I think that's interesting! That figure might be worth including – especially if your later results can explain it. It could be instructive about models.

Line 174: How sensitive are these NAO "spectral peaks" to the authors choice of dataset? How sensitive are they to the time period analyzed? I'll admit, my understanding of the spectral properties of the NAO index is heavily influenced by Wunsch (1999) – so my guess is that the red bands in Fig 3a will move around a bit. If they do move, it is worth asking, how do the authors' results change for different filter cutoffs?

Figures: I found the figures a bit too small to see details. If possible, please increase the size/resolution of the images. Thanks!

Figure 6: Is this a pixel-wise regression or a regression that takes place at each grid point? I assume, but please clarify in-text.

Figure 8a: Same as Figure 6.

Technical Comments:

Line 16: Please clarify the word "obvious"

Lines 21-23: For readers of the abstract, please clarify whether the authors are referring to "meridional advection" by ocean currents (e.g. Ekman) or winds (e.g. southerly component alongside eastern North America).

Line 28: I don't think I would consider the NAO an "event".

Line 40: I recommend changing "period" to "phase" since so much analysis of the NAO takes place in frequency-space

Lines 59 – 60: Please clarify the sentence containing "CMIP5-ESMs"

Line 88: You refer to "sea water Y velocity data" as "sea-surface meridional velocity" later in the paper. I think meridional velocity is a bit more clear – I would recommend using throughout.

Lines 101 – 104: Please clarify where the stations are for the station-based NAO index.

Lines 116 – 117: Can the authors please clarify this sentence. From this sentence alone, I can't understand how "8 year" periods and the "decadal" NAO go together. Interesting paper, though!

Line 154: I think "biased" might be the wrong word here, since the authors are comparing observations to observations.

Lines 254 - 258: This sentence is a bit long an awkward.

Lines 268 – 272: Again, the sentence here is distracting from the useful analysis.

Works Cited

Buckley, M.W., and J. Marshall. "Observations, Inferences, and Mechanisms of the Atlantic Meridional Overturning Circulation: A Review." Reviews of Geophysics 54, no. 1 (2016): 5–63. https://doi.org/10.1002/2015RG000493.

Cane, Mark A., Amy C. Clement, Lisa N. Murphy, and Katinka Bellomo. "Low-Pass Filtering, Heat Flux, and Atlantic Multidecadal Variability." Journal of Climate 30, no. 18

(2017): 7529–7553.

Delworth, Thomas L., Fanrong Zeng, Liping Zhang, Rong Zhang, Gabriel A. Vecchi, and Xiaosong Yang. "The Central Role of Ocean Dynamics in Connecting the North Atlantic Oscillation to the Extratropical Component of the Atlantic Multidecadal Oscillation." Journal of Climate 30, no. 10 (February 23, 2017): 3789–3805. https://doi.org/10.1175/JCLI-D-16-0358.1.

Deser, Clara, Michael A. Alexander, Shang-Ping Xie, and Adam S. Phillips. "Sea Surface Temperature Variability: Patterns and Mechanisms." Annual Review of Marine Science 2 (2010): 115–143.

Kay, J. E., C. Deser, A. Phillips, A. Mai, C. Hannay, G. Strand, J. M. Arblaster, et al. "The Community Earth System Model (CESM) Large Ensemble Project: A Community Resource for Studying Climate Change in the Presence of Internal Climate Variability." Bulletin of the American Meteorological Society 96, no. 8 (2015): 1333–49. https://doi.org/10.1175/BAMS-D-13-00255.1.

Wang, Chunzai, Liping Zhang, Sang-Ki Lee, Lixin Wu, and Carlos R. Mechoso. "A Global Perspective on CMIP5 Climate Model Biases." Nature Climate Change 4, no. 3 (March 2014): 201–5. https://doi.org/10.1038/nclimate2118.

Wunsch, Carl. "The Interpretation of Short Climate Records, with Comments on the North Atlantic and Southern Oscillations." Bulletin of the American Meteorological Society 80, no. 2 (1999): 245–255.

---

## Author Comment (AC1) · 4 Aug 2020

We would like to thank the reviewers for their time and efforts spent on reviewing our manuscript, and giving us good suggestions. We have carefully thought about these suggestions, and made a lot of changes to the article. Some new figures have been added, including some new analysis, and some inappropriate descriptions have been corrected. We have re-analyzed the influence of the NAO-driven SHF / LHF / surface seawater meridional velocity on SST and changed the time period of the analysis. We also have discussed the influences of the cutoff period of band-pass filter, different definitions of the NAO index, initial fields and external forcing of the historical experiments

on the results. All the changes are highlighted in the revised manuscript.

Responses to anonymous reviewer's comments point by point:

**Reviewer #2:**

**General Comments**

This manuscript shows that in several CMIP5 historical runs, the wintertime seasurface temperature (SST) response to the North Atlantic Oscillation (NAO) is not consistent with observations, on interannual timescales. The authors demonstrate that some models that exhibit this discrepancy fail to reproduce the observed relationship between SST and turbulent heat iňĆuxes (TFH), particularly in the subpolar gyre. They attribute this to an over-iniňĆuential ocean. Most models examined here correctly produce the interannual NAO-SST relationship in the subtropical Atlantic because of the larger iniňĆuence of Ekman forcing. 1 Do the results show model dependence or sensitivity to initial conditions? I think readers will find the results to be more convincing if the authors could rule out dependence on initial conditions in the NAO – SST relationship. One way of doing this might be to look at a single-model large ensemble (e.g. ; Kay et al. 2015). For unfiltered output, it seems like the authors' results should hold (http://webext.cgd.ucar.edu/Multi-Case/CVDP\_ex/cesm1.lens\_1920-2018/nao.tempreg.djf.png) – but is that still true when bandpass filtered?

Reply: Thank you for the suggestions. Kay et al. (2015) mentioned that the ocean initial conditions can influence multi-century coupled climate model runs, so whether the CMIP5-ESMs sensitivity to initial conditions is worth exploring. We have compared the relationship of the NAO and SST simulated by the models in different experiments which were initialed from different integrated time of piControl experiments (eg., r1i1p1 and r3i1p1) (P28-29, L481-496), and found that "but it should be emphasized that the influence of the initial conditions on the result needs to be considered in the evaluation of some individual models" (P29, L495-496)
2. If the results show model dependence, how do the different models responses to historical forcing influence your results? On line 321, the authors note that the NAO has a negligible response to shortwave radiation – but shortwave radiation does affect SST. Can we be sure that the SHF and LHF are not responding to externally-forced changes in SST over this twentieth century, with the NAO being a bystander? In other words, if the models are responding in different ways to historical forcing, is that alone enough to change the relationship between the NAO and SST/SHF/LHF? One way to address this question may be to look at pre-industrial control runs of the same CMIP5 models the authors already examine.

Reply:The relationships of the NAO and SST / SHF / LHF simulated by pre-industrial control runs of the 13 CMIP5-ESMs are analyzed (Figs. S13-14). The results of picontrol experiments by the same models are very similar with those of historical experiment (r1i1p1), but there are some differences in the intensity of the response of the SST/SHF/LHF to the NAO between different experiments. We have also added this part to the discussion section. (P29-30, L497-510)

3. Can the authors make causal claims based on band-pass filtered output? Cane et al. (2017) show that it is difficult to make causal claims about the sign of the relationship between heat fluxes and SST based solely on low-pass filtered data. I think that to show the causal relationship that is described in-text, it would be useful to also present unfiltered/annual average/wintertime average plots so that readers can be sure that the relationships shown are not an artifact of filtering. If the unfiltered results are similar, Ithink that is worth mentioning in-text.

Reply: Thank you for the reviewer's suggestion. We have done regression analysis of unfiltered winter average SST anomalies and NAO indexes / NAO-driven SHF / LHF anomalies and found that the patterns of observed and simulated unfiltered relationship of the NAO and SST by most of models are consistent with the filtered results (Figs. S7-8). The difference between the filtered and unfiltered results is discussed in Section 5.2 (P25-27, L437-458).
Specific Comments:

Lines 41 - 45: The wording and structure here is a bit too close to Deser et al. 2010 (page 119). Please edit.

Reply: Following the reviewer's suggestion, we have rewritten those sentences. "During the positive phase of the NAO, the westerly winds in the subpolar NA and the northeast trade winds in the tropical NA are strengthened, which causes the increased turbulent heat flux from the ocean to the atmosphere, while in the middle latitudes of the NA wind speeds are weakened, which causes the reduced turbulent heat flux out of the ocean" (P3, L37-40).

Line 46 – 48: I would recommend re-wording to emphasize that the "uniform SST warming" occurs after the NAO+ in observations and pre-industrial control runs of climate models (see Delworth et al. 2017). Further, I would caution that this mechanism is not established as causal in observations (e.g. Buckley and Marshall 2016).

Reply: Thanks for the reviewer's recommendation. We have rewritten these sentences to "After the positive phase of the NAO, some studies based on models suggest that, the Atlantic meridional overturning circulation (AMOC) is intensified, and the strengthened meridional heat transport associated with enhanced AMOC leads to broad scale SST warming (Sun et al., 2015, Delworth et al., 2017). Compared with other seasons, this phenomenon is more obvious in winter (Flatau et al., 2003; Bellucci et al., 2006), and probably occurs on the interdecadal and multidecadal scales (Eden and Jung, 2001, Gastineau et al., 2012). It should be noted that because there is a lack of long-time AMOC observations and the AMOC plays a more active influence on the change of SST on the long timescale (interdecadal and multidecadal scales), observational studies have not successfully linked the SST changes to the AMOC variability (Buckley and Marshall 2016)." (P3, L41-48)

Sun, C., Li, J. P., and Jin, F. F.: A delayed oscillator model for the quasiperiodic multidecadal variability of the NAO, Clim. Dyn., 45(7), 2083-2099, Interactive comment

https://doi.org/10.1146/10.1007/s00382-014-2459-z, 2015.

Delworth, T. L., Zeng, F. R., Zhang, L. P., Zhang, R., Vecchi, G. A., and Yang, X. S.: The central role of ocean dynamics in connecting the North Atlantic Oscillation to the extratropical component of the Atlantic Multidecadal Oscillation, J. Clim., 30(10), 3789–3805, https://doi.org/10.1175/JCLI-D-16-0358, 2017.

Flatau, M. K., Talley, L., and Niiler, P. P.: The North Atlantic Oscillation, surface current velocities, and SST changes in the subpolar North Atlantic, J. Clim., 2355-2369, https://doi.org/10.1175/2787.1, 2003.

Bellucci, A., and Richards, K. J.: Effects of NAO variability on the North Atlantic Ocean circulation, Geophys. Res. Lett., 33(2), L02612, https://doi.org/10.1029/2005gl024890, 2006.

Eden, C., and Jung, T.: North Atlantic interdecadal variability: oceanic response to the North Atlantic Oscillation (1865-1997), J. Clim., 14(5), 676-691, https://doi.org/10.1175/1520-0442(2001)0142.0.CO;2, 2001.

Gastineau, G., D'Andrea, F., Frankignoul, C.: Atmospheric response to the North Atlantic Ocean variability on seasonal to decadal time scales, Clim. Dyn., 40(9-10), 2311-2330, https://doi.org/10.1007/s00382-012-1333-0, 2012.

Line 64: Please clarify that Wang et al. (2014) found the NA SST was "underestimated" and "unreasonable" relative to observations (i.e. biased).

Reply: We have clarified the content in the revised version: "Wang et al. (2014) evaluated the global SST simulated by the CMIP5 models and found that the SST in the Northern Hemisphere, especially in the NA, is underestimated relative to the observation, and pointed out that it is mainly caused by the weaker AMOC and shallower AMOC cell compared to the observations.". (P4, L63-66)

Lines 157 – 160: So, the NAO centers of action are stable with time in models, but not observations? I think that's interesting! That figure might be worth including –especially

OSD
if your later results can explain it. It could be instructive about models.

Reply: This figure has been added to the supplement (Fig. S1). We regret that we haven't found the cause of this situation, so we have not added the figure in main text. We will continue to study this in the future.

How sensitive are these NAO "spectral peaks" to the authors choice of dataset? How sensitive are they to the time period analyzed? I'll admit, my understanding of the spectral properties of the NAO index is heavily influenced by Wunsch (1999) – so my guess is that the red bands in Fig 3a will move around a bit. If they do move, it is worth asking, how do the authors' results change for different filter cutoffs?

Reply: We have tested the sensibility of the periods of the NAO indexes to dataset and the time period analyzed in Jing et al., (2019). For example, the period of the NAO index provided by NOAA in 1950-2017 is 2.3-2.7 and 6 years, and that provided by NCAR in 1865-2017 is 2.3–2.7, 4.5 and 8.3 years. In this manuscript the period of the NAO index provided by NCAR from 1965 to 2015 are 3, 4.8, and 8-10. Therefore, the period of the NAO index is slightly different in different dataset or time range. According to this, we agree with the reviewer's suggestion to test how the results change with different filter cutoffs. We have done regression analysis of winter average SST anomalies and NAO indexes on the interannual scale calculated by 2-4 year filtering , and found that the observed and simulated values of the RCs and their patterns of the SST against the NAO based on 2-4 year filtering are close to the results based on 2-6 year filtering, so we think that the influence of the cutoff period used in the filter can be ignored (Fig. S9). (P27, L459-470)

Figures: I found the figures a bit too small to see details. If possible, please increase the size/resolution of the images. Thanks!

Reply: Sorry about that. We have improved the resolution of the figures, and some information such as spatial correlation coefficient and root mean square error has been enlarged.

OSD
Figure 6: Is this a pixel-wise regression or a regression that takes place at each grid point? I assume, but please clarify in-text. Figure 8a: Same as Figure 6

Reply: Yes, the regression takes place at each grid. We add a sentence "The regression and covariance are performed with the standardized variables, for which the regression is conducted at each grid." (P7, L109-110)

Technical Comments:

Line 16: Please clarify the word "obvious".

Reply: We have rewritten the abstract, so this sentence was deleted.

Lines 21-23: For readers of the abstract, please clarify whether the authors are referring to "meridional advection" by ocean currents (e.g. Ekman) or winds (e.g. southerly component alongside eastern North America).

Reply: Done. The meridional advection means ocean currents.

Line 28: I don't think I would consider the NAO an "event".

Reply: Thanks for the reminder. We have rewritten this sentence to "There is a strong inverse relationship between Iceland's and the Azores' monthly mean sea level pressure (most significant in winter) in the North Atlantic (NA), which is called the North Atlantic Oscillation (NAO) (Walker, 1924)" (P2, L26-27)

Line 40: I recommend changing "period" to "phase" since so much analysis of the NAO takes place in frequency-space.

Reply: Done.

Lines 59 - 60: Please clarify the sentence containing "CMIP5-ESMs".

Reply: The CMIP5-ESMs means the CMIP5 Earth System Models. We have changed "CMIP5-ESMs" to "CMIP5 models"

Line 88: You refer to "sea water Y velocity data" as "sea-surface meridional velocity"
later in the paper. I think meridional velocity is a bit more clear – I would recommend using throughout.

Reply: We use "sea-surface meridional velocity" throughout in the revision.

Lines 101 – 104: Please clarify where the stations are for the station-based NAO index.

Reply: We have clarified in section 2.2. 'The site-based observation-based NAO index is the difference of standardized sea level pressure between Lisbon, Portugal and Stykkisholmur / Reykjavik, Iceland' (P7, L111-113)

Lines 116 – 117: Can the authors please clarify this sentence. From this sentence alone, I can't understand how "8 year" periods and the "decadal" NAO go together. Interesting paper, though!

Reply: We have changed the analysis period, so the period of the NAO has changed. In this revised version, the observation-based NAO has the period of 8-10 years. It is generally believed that the period of 8 years or longer time can be treated as decadal signals.

Line 154: I think "biased" might be the wrong word here, since the authors are comparing observations to observations.

Reply: We have changed "biased" to "shift", "this shift is related to the phase of the NAO" (P11, L183)

Lines 254 - 258: This sentence is a bit long an awkward.

Reply: We've rewritten this sentence: "There may be two reasons for the bias of the locations and magnitude of negative response centers of the winter-averaged SST anomalies to the NAO-driven SHF anomalies by models: The areas where air-sea interaction is dominated by the atmosphere are different from the observation; there may be other factors which play dominate role to the variation of the SST and further impact the relationship between the anomalies of the SST and SHF".(P19, L321-324)
Lines 268 – 272: Again, the sentence here is distracting from the useful analysis.

Reply: Sorry about that. We've rewritten this sentence: "When the SST anomalies lags by 2 months onto SHF anomalies, all CMIP5 models can reproduce the negative covariance between SHF and SST anomalies in the most regions of the NA, although there are some models that simulate weak positive covariance in some regions of the subpolar NA, such as GFDL-ESM2M, HadGEM2-CC/ES, IPSL-CM5A-L/MR, MPI-ESM-LR, and MRI-ESM1, indicating that other factors (such as the internal motion of ocean) have an impact on the variations of the SST in the regions beyond the SHF in these models."(P19-18, L332-336)

---

## Author Comment (AC2) · 4 Aug 2020

We would like to thank the reviewers for their time and efforts spent on reviewing our manuscript, and giving us good suggestions. We have carefully thought about these suggestions, and made a lot of changes to the article. Some new figures have been added, including some new analysis, and some inappropriate descriptions have been corrected. We have re-analyzed the influence of the NAO-driven SHF / LHF / surface seawater meridional velocity on SST and changed the time period of the analysis. In addition, we have discussed the influences of the cutoff period of band-pass filter, different definitions of the NAO index, initial fields and external forcing of the historical

experiments on the results. All the changes are highlighted in the revised manuscript.

Responses to two reviewers' comments point by point:

Reviewer #1: The manuscript addresses the response of winter SST in the North Atlantic to NAO forcing in 13 CMIP5 models. Patterns of observation and models are compared. Because the patterns of NAO-driven SSTs look different between models and observation, the manuscript further investigates the link between the surface heat fluxes and the SST in the North Atlantic. Also, the link between the meridional ocean velocity and the NAO is presented. The authors suggest that it is the overestimated role of the ocean that causes an unrealistic relation between heat fluxes and SST, that finally cause the model to simulate a NAO-driven SST response that differs from the observed pattern. The basic motivation for the authors of this study is that the observed tripolar SST response to the NAO is simulated only by 7 models (as they write). However. I would say that more than only 7 model reproduce a tripolar pattern associated with the NAO on this timescale, even though the centres may be partly displaced, or not providing the correct amplitude (Fig. 4). I agree that there are models that cannot reproduce a realistic pattern, but most of them simulate a tripolar pattern. Especially, the positive centre near the American coast, is reproduced by all models. So, for me the research question would rather be, why are the subpolar (negative) centres displaced.I would hypothesize, that the displacement of the location could be explained by: the wrong location of the NAO-driven heat flux forcing, a different mean ocean circulation in the models, a different response of the circulation to the NAO, or a combination of these aspects.

Reply: We agree with the reviewer. The simulated NAO-SST relationship by CMIP5 has been re-described in the revised version of the manuscript. "Most of the models can reproduce an observed tripolar pattern of the response of the SST anomalies to the NAO on the interannual scale. The model bias is mainly reflected in the locations of the negative response centers in the subpolar NA (45-65°N), which is mainly caused by the bias of the response of the SST anomalies to the NAO-driven turbulent heat flux (THF)
anomalies."(Abstract, L13-16) "Compared with the observation, most of the models can roughly reproduce the tripole pattern of the response of the SST anomalies to the NAO. In the region around 20°N, all models can reproduce the significant negative response (reaching a 95% confidence level) east of 40°W, and in the subtropical NA, 10 models can reproduce significant positive response of the SST anomalies to the NAO near the American coast. The main difference in the RCs between the modeled and observation-based results occurs in the subpolar region where the simulated locations of the negative response centers by some of models are different from the observation-based results, especially in CanESM2, HadGEM2-ES, IPSL-CM5A-MR, MPI-ESM-LR / MR, and NorESM1-MR." (P14, L230-236)

The authors suggest that the cause for the unrealistic SST response is the incorrect response of SST to heat flux forcing, or as they write in the abstract for the subpolar North Atlantic 'most of the models simulate a positive response of SST to the turbulent heat flux'. And here I see a fundamental problem: When positive flux anomalies (ocean to atmosphere) are correlated with positive SST anomalies, then the SST is the driver for this link, not vice versa. Therefore, the regression that their conclusions are based on (Fig. 8 / 9), do simply not reflect the 'response of SST to the NAO via heat flux forcing'. The regressions seem, instead, to pick up something else, which may or may not be indirectly related to the NAO (for example through an ocean feedback). It could be that for some models / regions on the analysed timescales the dominant link between SST and heat fluxes is not the NAO-heat fluxes forcing the SST. To really extract the response of SST to a pattern of the heat fluxes that has been shown to be is NAO-driven for each model (maybe an index representing the typical structure as seen in Fig. 7).

Reply: We have revised this part of the analysis (sections 3.2.2 and 3.2.3). We used the least-squares method to extract NAO signals from heat fluxes, and then calculate the regression coefficient of the standardized SST anomalies against the standardized

**OSD**
NAO-driven heat flux (Figs. 8a and 9a). In the subpolar region the locations of negative RCs of the SST anomalies against NAO-driven heat flux anomalies (Figs. 8a and 9a) are very consistent with that of the SST anomalies against the NAO by most models (Fig. 4). This demonstrates that the bias of the response of the SST to the NAO-driven heat flux in models can influence the NAO-SST relationship in the region.

In summary, I see a fundamental problem with the interpretation of the results and based on that also not enough evidence for the conclusions presented here. Another issue is, that the motivation for this study are the differences in the SST response to the NAO (Fig. 4). But already within the same SST dataset, there are differences depending on how the NAO index was calculated (the first two panels in the first row of Fig. 4). Next, if I understood correctly, the NAO index in the models is calculated with another (third) method. So, it can be assumed that a part of the differences is explained by how the NAO index was calculated.

Reply: Thanks for the reviewer's suggestion. In the revised version, we used only one site-based NAO index based on the observation, which is provided by NCAR. However, we tested the effect of the different NAO index on the NAO-SST relationship in the section of discussion (section 5.3, P27-28, L472-479). The observation-based two NAO indexes defined by Gong and Wang (2000)'s method and the method used to calculate model NAO indexes (Zheng et al., 2013) are employed to study the response of the SST to the NAO on the interannual scale (Fig. S10). It is found that the definitions of the NAO index indeed affect the relationship between the NAO and SST in the subtropical NA, but have little impact in most regions of the subpolar and subtropical NA.

Furthermore, the entire manuscript would need substantial improvements regarding grammar / language in general. Therefore, I didn't list all the language issues or unclear formulations, because there were just too many. Based mainly on the concern that I have regarding the approach / interpretation of the results, I cannot recommend this manuscript for publication.
Reply: Sorry about that. We have checked the article carefully and corrected some mistakes in grammar / language.

Specific comments:

I suggest to modify the title to not have 'CMIP5 models on the interannual scale' together. So maybe 'Assessment of responses of North Atlantic winter SST to the NAO on the interannual scale in 13 CMIP5 models'.

Reply: Taking the reviewer's advice, we have changed the title.

15: Please clarify on the word 'obvious', in observations or models?

Reply: It means that the influences of sensible / latent heat fluxes on SST are both important in observation. In the revised version, the abstract has been rewritten, and this word is deleted.

20: For the sub-tropical region an 'incorrect positive response' is mentioned. Why is it 'incorrect' when this subtropical centre of the tripolar pattern should be positive? Further down it is also written 'models can simulate the realistic positive response of SST anomalies to the NAO in the subtropical NA'. which seems to be a contradiction.

Reply: The previous 'incorrect positive response' refers to the response of the SST to the LHF, and the latter positive response refers to the response of the SST to the NAO. We have rewritten the abstract in the revised version, and this sentence is deleted.

Overall, I find the abstract hard to understand.

Reply: We've rewritten the abstract.

The timescales are not made clear in the abstract.

Reply: We emphasized the timescales in the abstract of the revision (P1, L12-13).

59/60: 'In recent years, more and more people have realized that the evaluation of the CMIP5 Earth System Models (CMIP5-ESMs) is the basis for study by these models.' I
am not sure what the authors are trying to say here?

Reply: Sorry about this, we have changed the sentence to "In recent years, more and more people have realized that the identification of the CMIP5 Earth System Models bias is important for the improvement of these models and development of climate forecast (Wang et al., 2014, Wang et al., 2014)." (P4, L59-61)

Wang, G., Dommenget, D. and Frauen, C.: An evaluation of the CMIP3 and CMIP5 simulations in their skill of simulating the spatial structure of SST variability, Climate Dynamics, 44(1-2), 95-114, https://doi.org/ 10.1007/s00382-014-2154-0, 2014

Wang, C. Z, Zhang, L. P., Lee, S. K., Wu, L. X., and Mechoso, C. R.: A global perspective on CMIP5 climate model biase, Nat. Clim. Change, 4(3), 201-205, https://doi.org/10.1038/nclimate2118, 2014.

64/65: 'unreasonable simulation of AMOC'. In which way unreasonable?

Reply: The 'unreasonable simulation of AMOC' refers to that the AMOC strength is weak and the AMOC cell is shallow. We change this sentence to "it is mainly caused by the weaker AMOC and shallower AMOC cell compared to the observations." (P4, L65-66)

Why were these 13 models chosen? I would assume that SST and heat fluxes are widely available across CMIP5 models. Still, it seems that even out of this 13, two do not provide wind speed (Fig. 5 and 6).

Reply: We analyzed the response of North Atlantic SST to the NAO in order to analyze the response of the North Atlantic air-sea exchange carbon fluxes to the NAO in the future, because the SST plays an important influence on the air-sea carbon fluxes. Our research group has previously analyzed the fluctuation of global air-sea carbon fluxes on the interannual scale (Dong et al., 2016), and found 14 models with relatively rich outputs of biochemical variables from 22 cmip5 models (Dong et al., 2017). We would like to continue to analyze these 14 models in this paper, but unfortunately, we missed

**OSD**
one.

Dong, F., Li, Y., C., Wang, B., Huang, W., Shi, Y. Y., and Dong, W.: Global air-sea CO2 flux in 22 CMIP5 models: multi-year mean and interannual variability, J. Clim., 29(7), 2407-2431, https://doi.org/10.1175/JCLI-D-14-00788.1, 2016.

Dong, F., Li, Y. C., and Wang, B.: Assessment of responses of tropical Pacific air-sea CO2 flux to ENSO in 14 CMIP5 models. J. Clim., 30(21), 8595-8613, https://doi.org/ 10.1175/JCLI-D-16-0543.1, 2017.

What has been done with trends in the data, especially, when computing regressions?

Reply: The trends of the data in this study are removed by the least square method. We use the detrended data to obtain the standardized data, and then to compute regressions. (P7, L107-110)

Why do the 'regression coefficients' not have units? Are we actually looking at correlation coefficients here? What about the units for the covariances shown in Fig. 8b and 9b? What about the units for the covariances shown in Fig. 8b and 9b? 145: How exactly 'normalized'?

Reply: The 'normalized' should be 'standardized', and we have modified this word in the revision. The standardized data were calculated by dividing the anomalies (the trend is subtracted) of these variables by the standard deviation of these anomalies. (P7, L107-110). The regression analysis in this study was conducted with these standardized data, so the regression coefficients have no units. We also used the standardized data to calculate the covariances, so the covariances have no units too.

Is heat flux computed manually or is it a model output? And are the heat flux measures of observations and models derived /computed in a consistent way?

Reply: The modeled heat fluxes are model outputs. The observational heat flux is obtained directly from NOAA-CIRES, which is the reanalysis data and is the result of the 20CR that utilizes an Ensemble Kalman Filter data assimilation system. We explain
their possible differences in the revision. (P9, L155-158)

100: Please explained how the 'site-based' index is computed. Because it causes different regressions patterns (as seen in Fig. 4).

Reply: The site-based NAO index is the difference of normalized sea level pressure between Lisbon, Portugal and Stykkisholmur / Reykjavik, Iceland since 1864. (NCAR; www.climatedataguide.ucar.edu/climate-data/hurrell-north-atlantic-oscillationnao-index-station-based, Hurrell and Deser, 2009). (P7, L111-113)

Hurrell, J. W., and Deser C.: North Atlantic climate variability: the role of the North Atlantic Oscillation, J. Marine Syst., 79(3), 231-244, https://doi.org/10.1016/j.jmarsys.2009.11.002, 2009.

157: I don't understand this sentence: 'Because the locations of the NAO action centers simulated by most of the CMIP5 ESMs in different NAO phases do not show the movements illustrated by the observation (the figure is omitted), the differences between the models are not caused by the NAO period or the phase of the initial sign, but are only related to the structures of models.' So in observations the NAO pattern is not symmetric? And in the models it is symmetric? Or are the patterns also not symmetric, but differing from the observations? Also, I would argue that it is enough to say that - given the long period of 108 years and the rather short-timescale behaviour of the NAO - there is no reason to think the initial state would matter. But maybe I understood wrong what this sections was supposed to say.

Reply: In the observation, the NAO pattern is not symmetric (Fig. S1 in the revised version), but the NAO pattern simulated by most models is symmetric. The un-symmetric NAO pattern based on the observation has been shown by many studies (Jung et al. 2003; Moore, Renfrew, and Pickart 2013). The shift of the NAO action center is related to the phase of the NAO (Cassou et al. 2004, Jing et al., 2019). The location of the NAO action centers simulated by CMIP5 models in the different phases of the NAO does not move obviously in most models (Fig. S1 in the revised version). The
initial fields can influence the relationship of the NAO and SST (section 5.4), so we have revised the sentence to "These models are forced by the same external-forcing data. Therefore, the differences between the NAO patterns simulated by these models may be probably induced by their different structures and values of parameters." (P11, L180-182)

Jung, T., Hilmer, M., Ruprecht, E., Kleppek, S., Gulev, S. K., and Zolina. O.: Characteristics of the Recent Eastward Shift of Interannual NAO Variability, Journal of Climate, 16 (20), 3371–3382, doi:10.1175/1520-0442(2003)016

version), and this paragraph has been rewritten (P12-13, L206-217).

185: 'Based on the above analysis, simulated periods of the NAO indexes and areaaveraged SST anomalies on the decadal scale are different from the results of observation.' I don't see the data that clearly support this statement. It would be helpful to see the individual power spectra, instead of only the periods of the peaks.

Reply: The individual power spectra of the NAO indexes and SST are shown in the supplement file (Figs. S3-5 in the revised version). We have rewritten this sentence as' Based on the above analysis, simulated periods of the NAO indexes on the interannual scale are more consistent with the results of observations compared to those on the decadal scale. The observed periods of the area-averaged SST in subtropical and subpolar NA only presents interannual signals.' (P13, L218-220).

187: 'mainly reflected [. . .] on the interannual scales'. But aren't Eden and Jung 2001 focusing on the inter-decadal scale of the SST response to the NAO and the role of ocean dynamics?

Reply: Eden and Jung (2001) mentioned that the SST does respond to the NAO on interannual and interdecadal scales, but the mechanisms on these two timescales are different. The oceanic response to interdecadal changes of the NAO is primarily driven by surface net heat flux variability, which impacts the SST through impacting the ocean circulation. This mechanism is different compared to the oceanic variability on interannual timescales, where surface heat flux and windstress variability can directly impact the relationship of the SST and the NAO.

Eden, C., and T. Jung. 2004. "North Atlantic interdecadal variability: oceanic response to the North Atlantic Oscillation (1865-1997)," J. Clim., 14(5), 676-691, https://doi.org/10.1175/1520-0442(2001)0142.0.CO;2, 2001.

Six models are named which have a positive response in the subpolar region. However, some of these have also an area with a negative response in the subpolar latitudes
(besides the positive one). So, if being generous with the exact location of the subpolar centre, nearly all models (maybe except for IPSL-CM5A-MR and MPI-ESMMR) show some kind of tripolar response to the NAO. Please comment on that.

Reply: We agree with the reviewer and think that although some models have small positive response centers which are inconsistent with the observation, most models can roughly reproduce the tripolar pattern of the response of the SST to the NAO. We have rewritten this paragraph.

This last sentence in this paragraph should be revised. Both GFDL models have a similarly strong positive centre as in the obs\_Gong panel. But again, I really recommend using the same method to compute the NAO index.

Reply: We have rewritten this paragraph and delete this sentence (P13-14, L226-238). Taking the reviewer's advice, we only analyze the observed NAO index provided by NCAR, because this set of NAO index has been recognized and widely used. The locations of the NAO action center simulated by different models are different, so we use another method to avoid the impact of the fixed-location based NAO index on the results (described in P7, L114-117), which is widely used to define the NAO index of CMIP5 model simulation (Zheng et al., 2013; Wang et al., 2017). The influence of different definitions of the NAO index on our results is also discussed in this revision (section 5.3, P27-28, L472-479).

215: Is the SHF and LHF computed or is it model output? This is not clear, because in 130 it is only write 'usually calculated' and equations are provided. Based on that, are the observational heat flux data obtained in the same way?

Reply: The simulated SHF and LHF are the model outputs, which has been described in Section 2.1. The observational heat flux is obtained directly from NOAA-CIRES, which is the reanalysis data and the result of the 20CR that utilizes an Ensemble Kalman Filter data assimilation system. We have added some explanation about these questions: "The methods adopted by the observation-based products and models to OSD
calculate the SHF / LHF are similar, which are mainly based on the bulk formula, but may use different parameters, so the above equations (2-6) only help us to understand the relationship between the SST and SHF / LHF, which are not the exact formulas used in the observation-based products and models."(P9, L155-158)

219 / Fig. S2: I cannot agree on the statement that all models overestimate the SHF north of 50N. First of all, the observations do not cover the area in the Labrador Sea, which seems to be the area of maximum ocean heat loss in the observations, which also seems to be the case for some of the models. So, I would say that, for example, both MPI models are doing quite well in reproducing the observed heat flux. Next, it is interesting that specifically the IPSL-CM5B-LR model is the one that is least 'overestimating'- I would rather say 'underestimating' the heat flux. And as mentioned before, I find this model least capable of reproducing the tripolar SST pattern associated with the NAO. In summary, I don't find it convincing that an overestimated heat flux (in the mean state) might be the cause for an unrealistic SST fingerprint in the models.

Reply: We agree with the reviewers. Most models seem to reproduce the maximum SHF in the Labrador Sea, which is consistent with the observation. Among them MPI-ESM-LR/MR are doing quite well in reproducing the observed SHF / LHF (Fig. S6 in the revised version). We did not consider the simulation deviation of the multi-year mean SHF and LHF as a factor affecting the NAO-SST relationship too. Fig. S6 only intends to show the direction of the heat fluxes. We have rewritten this paragraph (P16-17, L278-282). Č Another issue is, that I am not sure how robust the 'observations' regarding their heat flux mean states are (Fig. S2). I recommend to test that through showing only the heat fluxes during the last decades when higher quality and quantity of observations were available, and also showing the mean state of a different reanalysis product. It could be that they are indeed robust. It just needs to be shown, because the model performances are evaluated based on these results.

Reply: Thank the reviewer for the good suggestion. We chose 1965-2015 as the time period for analyzing observed multi-year mean SHF / LHF in the full text. The same
time interval of 1955-2005 was selected for the model data. In order to judge the adopted observed winter SHF / LHF's validity, three other reanalysis data were selected and compared with each other in the revised version. "In order to make sure the accuracy of the observed multi-year averaged winter SHF / LHF, three other observationbased SHF/LHF data in winter are selected and all of these datasets are in the same periods from 1980 to 2015. The distributions of the SHF / LHF from the 4 reanalysis databases are generally consistent with each other, and the main difference among these datasets is in the intensity of high values, especially in the high value center of the LHF located in the tropical NA (Figure 6a). The response of the SHF / LHF anomalies to the NAO in these 4 datasets is also close to each other, and the main difference among these datasets still occurs in the tropical NA (Figure 6b). Based on the above analysis, it can be concluded that the difference among the observation-based SHF / LHF does not affect the investigation of the relationship of the SHF / LHF and NAO / SST in this study because the regions of concern are mainly the subtropical and subpolar NA. In the following text, unless otherwise specified, the observation-based SHF / LHF is the data from the NOAA-CIRES 20th Century Reanalysis version 2." (P16, L268-277)

224: For SHF I would even say 30-65N. And also in the Gulf of Mexico and in the Caribbean. But again, it should be shown that these results based on the reanalysis product starting around the year 1900 are robust.

Reply: We agree with the reviewers. We use the reanalysis data from 1965 in the revision.

Fig. 6: I don't think Fig. 6 is useful. Naturally, increased wind speed tends to increase the heat flux (whether from ocean to atmosphere or atmosphere to ocean). When trying to explain the differences in the response of SST to the NAO, it would be more useful to compare the differences in quantities regressed onto the NAO index (like Figure 7), because as shown before (in Fig. 5), the wind-speed response to the NAO is non-uniform.

**OSD**
Reply: We agree with the reviewers. We've deleted Fig. 6. According to the direction of multi-year average SHF and LHF, it is described in the revision: Considering the directions of the SHF / LHF, the increase in wind speed can significantly increase the turbulent heat flux transported from a large region of sea surface to the atmosphere. (P16, L266-267)

238: Please also comment on the comparison of models. Fig. 8a: In the models with unrealistic positive correlations, is the atmospheric forcing (NAO) maybe too weak compared to other models / observations? It might be like that - when I compare the explained variances from Fig. 1.

Reply: Thank the reviewer for the reminder. We carefully considered the reviewer's opinion. In the revision, we use the least-squares method to extract NAO signals from the heat fluxes, and then calculate the standardized regression coefficients of the SST anomalies against the NAO-driven heat fluxes (Figs. 8a and 9a). Based on this treatment, the climate variation unrelated to the NAO has little effect on the relationship between the SST and SHF / LHF / NAO.

310: 'NAO-driven SHF / LHF anomalies': Regression between SST and HF without a direct relation to NAO were shown. Therefore, it is not justified to say 'NAO-driven'. This could only be said if the regressions had been done on an index (e.g., PC-based) that is related to the NAO.

Reply: We have corrected that statement. We use the regression method to establish the SHF / LHF / sea-surface meridional velocity anomalies driven by the NAO, and then calculate the standardized regression coefficient of the SST anomalies against the NAO-driven SHF / LHF / sea-surface meridional velocity anomalies.

336-339: I cannot agree on this statement, because: Fig. 7 (for heat fluxes) is the analogous version to Fig. 10 (for the meridional ocean velocity). Both figures show that the models reproduce the observations. Based on that only it is not justified to say that the root for an unrealistic SST response to the NAO are the heat fluxes. Indeed,

OSD
the regressions of heat fluxes and SST in Fig. 8 and 9 show that the heat flux / SST relations are not realistic. But an analogous analysis for the meridional surface ocean velocity / SST is not presented. Even here there might be differences. And then one cannot say that unrealistic aspects in the SST response are caused only by the wrong heat flux response.

Reply: We agree with the reviewer, and re-analyze this part. We calculate the correlation coefficients of the standardized SST and the SST change that is induced by the change of meridional heat transport related to the NAO-driven meridional sea surface velocity (Fig. 10b). These correlation coefficients show that the NAO-driven meridional sea surface velocity has no obvious regular influence on the SST. (P25, L421-435)

348: 'Because there is a deviation between the simulated and observed periods of the NAO indexes / area averaged SST on the decadal scale.' What is meant by 'period' and 'deviation' here exactly? The most dominant timescales of variability from the power spectra?

Reply: The 'period' means significant periods obtained by the power spectra. The 'deviation' means that the simulated significant periods (obtained by the power spectra) characterized by decadal signal are different from those observation, and some models do not reproduce the significant periods characterized by decadal signal. We delete this sentence in the revision, and rewrite this paragraph (P23, L391-394).

363: 'LHF and SST is mainly related.' What is meant by 'mainly related'?

Reply: It means that the variation of SST is mainly affected by the variation of LHF. We rewrite the conclusion, so this sentence has been deleted.

364: When the response to LHF is 'unreasonably positive', how can contribute to a too weak positive response in the subtropical NA (as mentioned further above in I. 355)?

Reply: In original version, it means that during the positive phase of the NAO, the increase of the meridional heat transport leads to the increase of the SST in the sub-
tropical NA. This mechanism probably conceals the unreasonable impact of the heat turbulent flux anomalies on the SST anomalies in some models, so that the simulated response of the SST anomalies to the NAO by most of the models is weak positive in the subtropical NA. In the revised version, we think that this mechanism may be wrong and delete this statement.

366-367: 'have a significant positive response to the NAO'. This only applies to the subtropics, not the subpolar region.

Reply: Thank the reviewer for reminding us. Because we have rewritten the conclusion, this sentence has been deleted.

As a supplementary document already exists, I would suggest to also show the individual power spectra from which Figure 3 is derived. For the power spectra please also provide the information about the window that is used to compute them.

Reply: We have added the figure of the power spectra of the NAO indexes and the areaaveraged SST in the subtropical / subpolar regions to the supplementary document (Figs. S3-5).

Technical corrections:

28: First sentence: A 'relationship' is not an 'event'.

Reply: We changed this sentence to "There is a strong inverse relationship between Iceland's and the Azores' monthly mean sea level pressure (most significant in winter) in the North Atlantic (NA), which is called the North Atlantic Oscillation (NAO) (Walker, 1924)."

220: By 'MPI-ESM1' you probably mean MRI-ESM1?

Reply: Yes, we have corrected it.

352: 'along the meridian' - which one?

OSD
Reply: The regression coefficient of the SST and NAO indexes are along the meridional direction. We re-described it.

355: 'weaker positive responses'. I would add 'than observed'.

Reply: Thank you for the reviewer's suggestions. The sentence has been deleted in the revised version.

366: 'observed meridional velocities'. Please add the information that it is the surface ocean velocity.

Reply: Done

'Constant field value is 10' is not a good annotation for a panel where model data are not available.

Reply: We have removed it from the figure.

In the figures with subpanels for the different models, sometimes there is one, sometimes there are two observational panels. That moves the position of the model panels and it is hard to compare them across figures. I suggest to have the observations as the last panels, or leave the second panel position free for the case there is no second observation panel.

Reply: We redraw all the figures to make sure that one observation is at the top left, and the position of each model in these figures is fixed.

Please increase the resolution of Figs. 1 and 3.

Reply: We have increased the resolution of these two figures.

I think the SCCs in Figure 2 are not a very representative measure, because they hardly vary despite the model differences (as seen on the maps, or in the RMSE).

Reply: We agree with the reviewer. We have removed SCCs in Fig. S2.

Figure 3: Please explain the meaning of the horizontal lines / areas.
Reply: The horizontal lines / areas mean the significant periods of observation. We have added the description for them under the corresponding figures.

**OSD**

---

## Referee Report (RR1)

Jing et al. 2020 Second Review

This manuscript shows that in several CMIP5 models the wintertime sea-surface temperature response to the NAO in the North Atlantic subpolar gyre is not consistent with observations on interannual timescales. They attribute this model bias to and incorrect (or inconsistent) turbulent heat flux response to the NAO induced SST tripole.

Overall, I'm satisfied with the scientific content of the manuscript and recommend acceptance pending minor revisions. Further, I would highly recommend the authors seek out additional (perhaps professional) copy editing. I think this is a useful and timely paper, so I encourage this copy editing to make sure the authors' work is understood by the community at-large.

*Minor Revisions*

L59 – 61: I recommend cutting the phrase "In recent years, more and more people have realized" (and adjusting the rest of the sentence).

L63 – 66: The wording is slightly ambiguous here. As you point out on L187-189, Wang et al. (2014) found that *mean* SSTs were too cold. (Also, there are two references for Wang et al. 2014. Can you please differentiate in some way? In the second Wang et al. (2014) reference, I believe that there is a missing "s" in the word "biases" in the article title.)

L180-183. I think that both of these sentences are correct individually – but I am not following the logic between them. I would recommend cutting or moving the first sentence.

L189 – 192 and Figure 2: This reminds me of Siqueira and Kirtman 2016 who show a change in ocean resolution can change the location of atmospheric circulation anomalies (their Figure 3).

 Siqueira, L., and B. P. Kirtman (2016), Atlantic near-term climate variability and the role of a resolved Gulf Stream, Geophys. Res. Lett., 43, 3964–3972,doi:10.1002/2016GL068694.

L239: "are slightly [further] south than observations" or "are slightly south [of ] observations" (and again on L240 – 241).

L255: "abnormal" -> "anomalous"

L358 – 363: I recommend breaking this up into multiple sentences.

L438 – 443: I also recommend breaking this up into multiple sentences.

L443: It is true that the unfiltered timeseries should have more degrees of freedom, but I'm not sure why we would expect that to influence the magnitude of the regression coefficients in one direction (negative) over another (positive). Perhaps this is just removing the autocorrelation induced by the comparing two filtered timeseries?

L466: see previous comment.

L475 and L476: "indexes" -> "indices"

L481: I believe the CESM Large Ensemble is initialized with minute perturbations in atmospheric temperature. Given these are free running models, I suspect that the differences the authors find are a result of the different time histories of internal variability that result from different initial conditions – and not from ocean initialization (as in a forecast model). I encourage the authors to make this distinction, if they agree.

L505 – 506: I'm not sure how you draw this conclusion.

*Optional suggestion:*

Section 5.4:

First, I found this additional section very helpful. I know it was a lot of work, but I hope it will make your paper more impactful. One reason I suggested this analysis was related to Scaife and Smith (2018)'s "signal-to-noise paradox", wherein models produce NAOs that are more like observations than themselves. In other words, the signal-to-noise ratio in the NAO in climate models is unrealistically low. I suspect that the authors are finding something similar (through a very welcome mechanistic approach). It may be useful for the impact of this work to tie these ideas together here. I'll note that since the first version of this manuscript, Smith et al. (2020) have published a high-impact paper that claims to overcome the signal-to-noise paradox through a very large ensemble (169 members).

---

## Author Response (AR2)

**Oct. 10, 2020.**

**Manuscript Number: os-2020-16**.

We would like to thank the editor and reviewer for reviewing our manuscript and giving us good suggestions. According to these suggestions, we have corrected some mistakes in expression and added some discussion.

**Responses to the reviewer' comments point by point:**

L59-61: I recommend cutting the phrase "In recent years, more and more people have realized" (and adjusting the rest of the sentence).

Reply: We cut the phrase and modified this sentence into "The identification of the CMIP5 Earth System Models bias is important for the improvement of these models and development of climate projection." **(P4, L61-62)**

L63-66: The wording is slightly ambiguous here. As you point out on L187-189, Wang et al. (2014) found that mean SSTs were too cold. (Also, there are two references for Wang et al. 2014. Can you please differentiate in some way? (In the second Wang et al. (2014) reference, I believe that there is a missing "s" in the word "biases in the article title.)

Reply: Both of the two sentences on **L63-66** and **L187-189** in the revision1 are to say that CMIP5 models underestimate the annual mean SST in the North Atlantic. In this revision, we change this sentence to "Meanwhile, Wang et al. (2014b) evaluated the global annual mean SST simulated by the CMIP5 models and found that the SST in the Northern Hemisphere, especially in the NA, is underestimated." **(P4, L65-68)** In addition, we use "a, b" to distinguish the two references from Wang et al., and we have added "s" in the word "biase" in the article title by Wang et al. (2014b). Thank the reviewer for the reminder.

L180-183. I think that both of these sentences are correct individually – but I am not following the logic between them. I would recommend cutting or moving the first

sentence.

Reply: What we would like to express is that the external forcing will not cause differences between the NAO patterns simulated in these models, because these models are forced by the same external-forcing data. We've changed the sentence to "The differences between the NAO patterns simulated by these models with the same external-forcing data are probably induced by their different model structures and values of parameters." (P11, L182-184)

L189-192 and Figure 2: This reminds me of Siqueira and Kirtman 2016 who show a change in ocean resolution can change the location of atmospheric circulation anomalies (their Figure 3).

Siqueira, L., and B. P. Kirtman (2016), Atlantic near-term climate variability and the role of a resolved Gulf Stream, Geophys. Res. Lett., 43, 3964-3972,doi:10.1002/2016GL068694.

Reply: Taking the reviewer's advice, we have added some discussion: "With a climate system model, Siqueira and Kirtman (2016) found that the change of ocean component model resolution can change the simulated SST variabilities, locations of atmospheric circulation anomalies, and air-sea interactions in the North Atlantic. The change is induced by the impact of the resolution on the ocean dynamics, such as ocean fronts and eddies in the Gulf Stream which can be well resolved in the high resolution model with the horizontal resolution of $0.1° \times 0.1°$. Nevertheless, the highest horizontal resolution of these ocean component models used in this study is $0.4° \times 0.4$ ° (MPI-ESM-MR), and the comparison of MPI-ESM-LR and MPI-ESM-MR, both of which are from the same institution and with different ocean component model resolutions, shows that the SST variability in the Gulf Stream is not significantly different. This indicates that the resolution of these models is still not enough to investigate the SST variability in the Gulf Stream and may induce the deviation between the simulated SST variability and the observed one." (P11-12, L193-203)

L239: "are slightly [further] south than observations" or "are slightly south [of ] observations" (and again on L240-241).

Reply: We've corrected the sentence to "In HadGEM2-ES, the low-pressure action centers of the NAO are slightly further south than observations, and the negative response center of the SST to the NAO is also further south than observations" **(P15, L250-252)**

L255: "abnormal" -> "anomalous"

Reply: Done.

L358 – 363: I recommend breaking this up into multiple sentences.

Reply: We've changed the sentence to "The distributions of the RCs are similar to those of the SST anomalies against NAO-driven SHF anomalies in a large area of NA. The main difference between the response of the SST to the SHF and to the LHF is that the observed and modeled positive RCs of the SST anomalies against NAO-driven SHF anomalies in the eastern NA around 20°N do not occur in the regression of the SST anomalies against NAO-driven LHF anomalies. It indicates that the influence of the LHF on the SST probably controls the RCs of the SST anomalies against the NAO in this region." **(P22, L370-374)**

L438 – 443: I also recommend breaking this up into multiple sentences.

Reply: We've changed the sentence to "We also did regression analysis of unfiltered winter average SST anomalies and NAO indices (Fig. S7). It is found that except for the models of IPSL-CM5A-MR and MPI-ESM-L / MR, there is no obvious difference in the distribution of standardized RCs of the SST and NAO between the filtered and unfiltered results, and the main difference is that the RCs from the unfiltered data are slightly smaller than those from the filtered data in the subtropical NA (Fig. 4) of both the observation-based results and most of the modeled results." **(P26, L450-454)**

L443: It is true that the unfiltered timeseries should have more degrees of freedom, but

I'm not sure why we would expect that to influence the magnitude of the regression coefficients in one direction (negative) over another (positive). Perhaps this is just removing the autocorrelation induced by the comparing two filtered timeseries?

L466: see previous comment.

Reply: Thank for the reviewer's reminder. The difference between unfiltered and filtered results is more obvious in the subtropical NA, where the regression coefficients between the NAO and SST are positive. The change of the degree of freedom can't explain this phenomenon, and there are also many problems that can't be explained by degrees of freedom, such as the inconsistency between the observed data and the model results due to the influence of filtering in the tropical and subpolar regions, so we have deleted these sentences about the degree of freedom. Taking the reviewer's suggestion, we analyzed the effect of autocorrelation: we removed the autocorrelation from the unfiltered NAO and SST with the Cochrane-Orcutt method before the regression analysis, and found that the magnitude of regression coefficients (Fig. R1) are very consistent with that from the original data (Fig. S7 in this revision). Therefore, we think the autocorrelation is not an important factor that causes the difference between the unfiltered and filtered results. At present, we have not thought of any explanation for this phenomenon. In our follow-up research, we will pay more attention to explore this problem.

[Figure]

Figure R1 Standardized regression coefficients (removing the autocorrelation of the NAO and SST) of the winter-averaged SST anomalies against the NAO indexes (without data filtering). Shaded areas indicate that RCs are statistically significant at the 95% confidence level of the Student's t-test. The obs is the RCs of observed SST to the NAO indexes provided by NCAR. The time periods for the observation and models range from 1965 to 2015 and 1955 to 2005, respectively. The simulated results are based on historical experiment of CMIP5 (r1i1p1).

L475 and L476: "indexes" -> "indices"

Reply: Done.

L481: I believe the CESM Large Ensemble is initialized with minute perturbations in atmospheric temperature. Given these are free running models, I suspect that the differences the authors find are a result of the different time histories of internal variability that result from different initial conditions, and not from ocean initialization (as in a forecast model). I encourage the authors to make this distinction, if they agree.

Reply: Yes, we agree with the reviewer. We have modified the paragraph into: "Kay et al. (2015) did ensemble experiments by adding different minute perturbations to the atmosphere as initial conditions to study the internal variability. There are also some ensemble historical experiments in CMIP5 which are initialized with different initial conditions in 1850. The initial conditions of these ensemble members are from the

different integrated time of the piControl experiments, so these initial conditions represent the different time histories of internal variability. The relationship of the NAO and SST simulated by the models with above mentioned different initial fields (r1i1p1 and r3i1p1) are compared (Fig. S11)." **(P29, L493-498)**

L505 – 506: I'm not sure how you draw this conclusion.

Reply: After the initial field of historical experiment is provided, the piControl experiment will continue to integrate for 500 years, so the result of the piControl experiment may be more stable than that of historical experiment in terms of the internal variability. The result of the piControl experiments in MPI-ESM-MR is very similar with the historical experiments (r3i1p1), but is different from the historical experiments (r1i1p1). We infer that the initial fields of the historical experiments (r1i1p1) of this model may come from an early integrated time of the piControl experiment. Because we can't find the material to support our conclusion, we delete this sentence in this revision.

**Optional suggestion:**

Section 5.4:

First, I found this additional section very helpful. I know it was a lot of work, but I hope it will make your paper more impactful. One reason I suggested this analysis was related to Scaife and Smith (2018)'s "signal-to-noise paradox", wherein models produce NAOs that are more like observations than themselves. In other words, the signal-to-noise ratio in the NAO in climate models is unrealistically low. I suspect that the authors are finding something similar (through a very welcome mechanistic approach). It may be useful for the impact of this work to tie these ideas together here. I note that since the first version of this manuscript, Smith et al. (2020) have published a high-impact paper that claims to overcome the signal-to-noise paradox through a very large ensemble (169 members).

Reply: Thanks for the reviewer's suggestion. We have read these two references (Scaife

and Smith, 2018; Scaife et al., 2020), and learned a lot from them. They mentioned that the current climate models can predict observed climate variability, although the predictable signal of the climate variability is small, especially in the Atlantic Ocean, and the small predictable signal may arise from an underestimate in the strength of the response to external forcing (such as volcanic forcing, solar variability, and ozone depletion). In our manuscript, based on the comparison of the piControl and historical experiments, we also found that when the external forces are changed, in most models, the NAO-SST is not changed obviously. Scaife et al. (2020) also did a lot of work to overcome the question of low signal-to-noise ratios, which also has a lot of inspiration for us. Unfortunately, since we do not have enough work basis to utilize the results from different models, initial fields and forced fields, we can't carry out the research work in this field within the short term. Thus, we have only added a short discussion in this paper. "Some studies have shown that in the climate models, the amplitude of the response to the external forcing (such as volcanic forcing, solar variability, and ozone depletion) is weak, which leads to weak predictable signals in these models although these models can predict observed climate variability (Scaife and Smith et al. 2018). The weak predictable signals inhibit the estimation of forced climate variability in the Atlantic sector (Scaife and Smith et al. 2018). The weak influence of the external forcing on NAO-SST relationship was also found in the CMIP5 models in this work. Scaife et al. (2020) have argued that a large number of ensemble can overcome the signal-to-noise paradox, which probably provide a reference for the future application of CMIP models in the predications." **(P30-31, L522-528)**